# Kinetic modeling predicts a stimulatory role for ribosome collisions at elongation stall sites in bacteria

**Michael A Ferrin, Arvind R Subramaniam\***

Basic Sciences Division and Computational Biology Program of Public Health Sciences Division, Fred Hutchinson Cancer Research Center, Seattle, United States

**Abstract** Ribosome stalling on mRNAs can decrease protein expression. To decipher ribosome kinetics at stall sites, we induced ribosome stalling at specific codons by starving the bacterium *Escherichia coli* for the cognate amino acid. We measured protein synthesis rates from a reporter library of over 100 variants that encoded systematic perturbations of translation initiation rate, the number of stall sites, and the distance between stall sites. Our measurements are quantitatively inconsistent with two widely-used kinetic models for stalled ribosomes: ribosome traffic jams that block initiation, and abortive (premature) termination of stalled ribosomes. Rather, our measurements support a model in which collision with a trailing ribosome causes abortive termination of the stalled ribosome. In our computational analysis, ribosome collisions selectively stimulate abortive termination without fine-tuning of kinetic rate parameters at ribosome stall sites. We propose that ribosome collisions serve as a robust timer for translational quality control pathways to recognize stalled ribosomes.

**\*For correspondence:** rasi@ fredhutch.org

**Competing interests:** The authors declare that no competing interests exist.

## Introduction

Ribosomes move at an average speed of 3–20 codons per second during translation elongation *in vivo* (*Dalbow and Young, 1975*; *Bonven and Gulløv, 1979*; *Yan et al., 2016*). Since this rate is higher than the typical initiation rate of ribosomes on mRNAs [less than 1 s$^{-1}$ (*Yan et al., 2016*; *Kennell and Riezman, 1977*)], elongation is often assumed to not affect the expression level of most proteins. Nevertheless, the elongation rate of ribosomes can decrease significantly at specific locations on an mRNA due to low abundance of aminoacyl-tRNAs, inhibitory codon pairs or amino acid pairs, nascent peptides interacting strongly with the ribosome exit tunnel, or the presence of RNA-binding proteins (*Richter and Coller, 2015*). Ribosome profiling — the deep sequencing of ribosome-protected mRNA fragments — has enabled the identification of additional factors that induce slowing or stalling of ribosomes during elongation (*Ingolia et al., 2009*; *Ingolia, 2014*). An important question emerging from these studies is the extent to which ribosome stalling affects the expression of the encoded protein, since initiation might still be the slowest step during translation.

Several mechanistic models have been proposed to explain how ribosome stalling during elongation might affect the expression of the encoded protein. In the widely used traffic jam model (*MacDonald et al., 1968*), the duration of ribosome stalling is sufficiently long to induce a queue of trailing ribosomes extending to the start codon, thus decreasing the translation initiation rate. Evidence supporting this model has been found in the context of EF-P dependent polyproline stalls in *E. coli* (*Hersch et al., 2014*; *Woolstenhulme et al., 2015*), and rare-codon induced pausing in *E. coli* and yeast (*Mitarai et al., 2008*; *Chu et al., 2014*). In an alternate abortive termination model, ribosome stalling causes premature termination without synthesis of the full-length protein. This model is thought to underlie the action of various ribosome rescue factors in *E. coli* and yeast

(*Subramaniam et al., 2014*; *Choe et al., 2016*). Finally, ribosome stalling can also affect protein expression indirectly by altering mRNA stability (*Presnyak et al., 2015*; *Radhakrishnan et al., 2016*), co-translational protein folding (*Chaney and Clark, 2015*), or stress-response signaling (*Ishimura et al., 2016*).

Despite the experimental evidence supporting the above models, predicting the effect of ribosome stalling on protein levels has been challenging because of uncertainty in our knowledge of *in vivo* kinetic parameters such as the duration of ribosome stalling and the rate of abortive termination. Further, while we have a detailed understanding of the kinetic steps and structural changes that occur during the normal elongation cycle of the ribosome (*Wintermeyer et al., 2004*; *Voorhees and Ramakrishnan, 2013*; *Blanchard et al., 2004*), the 'off-pathway' events that occur at stalled ribosomes have been elucidated in only a few specific cases (*Neubauer et al., 2012*; *Muto et al., 2006*; *Shao et al., 2015*). Thus, development of complementary approaches, which can quantitatively constrain the *in vivo* kinetics of stalled ribosomes without precise knowledge of rate parameters, will be useful for bridging the gap between the growing list of ribosome stall sequences (*Ingolia, 2014*; *Woolstenhulme et al., 2013*; *Gamble et al., 2016*) and their effect on protein expression.

Here, we investigated the effect of ribosome stalling on protein expression using amino acid starvation in *E. coli* as an experimental model. In this system, we previously found that both ribosome traffic jams and abortive termination occur at a subset of codons cognate to the limiting amino acid (*Subramaniam et al., 2013*). Motivated by these observations, here we computationally modeled ribosome traffic jams and abortive termination with the goal of predicting their effect on protein expression. Even without precise knowledge of *in vivo* kinetic parameters, we found that these two processes give qualitatively different trends in protein expression when the initiation rate, the number of stall sites, and the distance between stall sites are systematically varied. Surprisingly, experimental measurements support a model in which traffic jams and abortive termination do not occur independent of one another; rather, collisions by trailing ribosomes stimulate abortive termination of the stalled ribosome. We find that this model is consistent with the absence of long ribosome queues in ribosome profiling measurements, and it naturally provides a mechanistic basis for the selectivity of abortive termination towards stalled ribosomes. While these conclusions are limited to the specific context of amino acid starvation in *E. coli*, the integrated approach developed in this work should be generally applicable to investigate other ribosome stalls in both bacteria and eukaryotes.

## Results

### Effect of ribosome stalling on measured protein level, mRNA level, and polysome occupancy

During starvation for single amino acids in *E. coli*, certain codons that are cognate to the limiting amino acid decrease protein expression, while the same codons have little or no effect during nutrient-rich growth (*Subramaniam et al., 2013*). For example, synonymously mutating seven CTG leucine codons in the yellow fluorescent protein gene (*yfp*) to CTA, CTC, or CTT reduced the synthesis rate of YFP 10–100 fold during leucine starvation (*Subramaniam et al., 2013*). Genome-wide ribosome profiling showed that ribosomes stall at CTA, CTC, and CTT codons during leucine starvation, which leads to a traffic jam of trailing ribosomes and abortive termination of translation (*Subramaniam et al., 2014*). These observations led us to ask whether ribosome traffic jams and abortive termination can quantitatively account for the decrease in protein synthesis rate (number of full proteins produced per unit time) caused by ribosome stalling during leucine starvation in *E. coli*.

To measure the effect of ribosome stalling on protein synthesis during leucine starvation, we constructed fluorescent reporter genes which have a stall-inducing CTA codon at one or two different locations along *yfp* (*Figure 1A*, blue bars). We induced these reporter variants from very low copy vectors (SC*101 *ori*, 3–4 copies per cell) either during leucine starvation or during leucine-rich growth. While YFP expression was similar across all *yfp* variants during leucine-rich growth, a single CTA codon at two different locations reduced YFP expression during leucine starvation by 3–4 fold relative to a control *yfp* without CTA codons (*Figure 1B*). Introducing both CTA codons reduced YFP expression by ~6 fold (*Figure 1B*), and a stretch of 7 CTA codons reduced YFP expression close

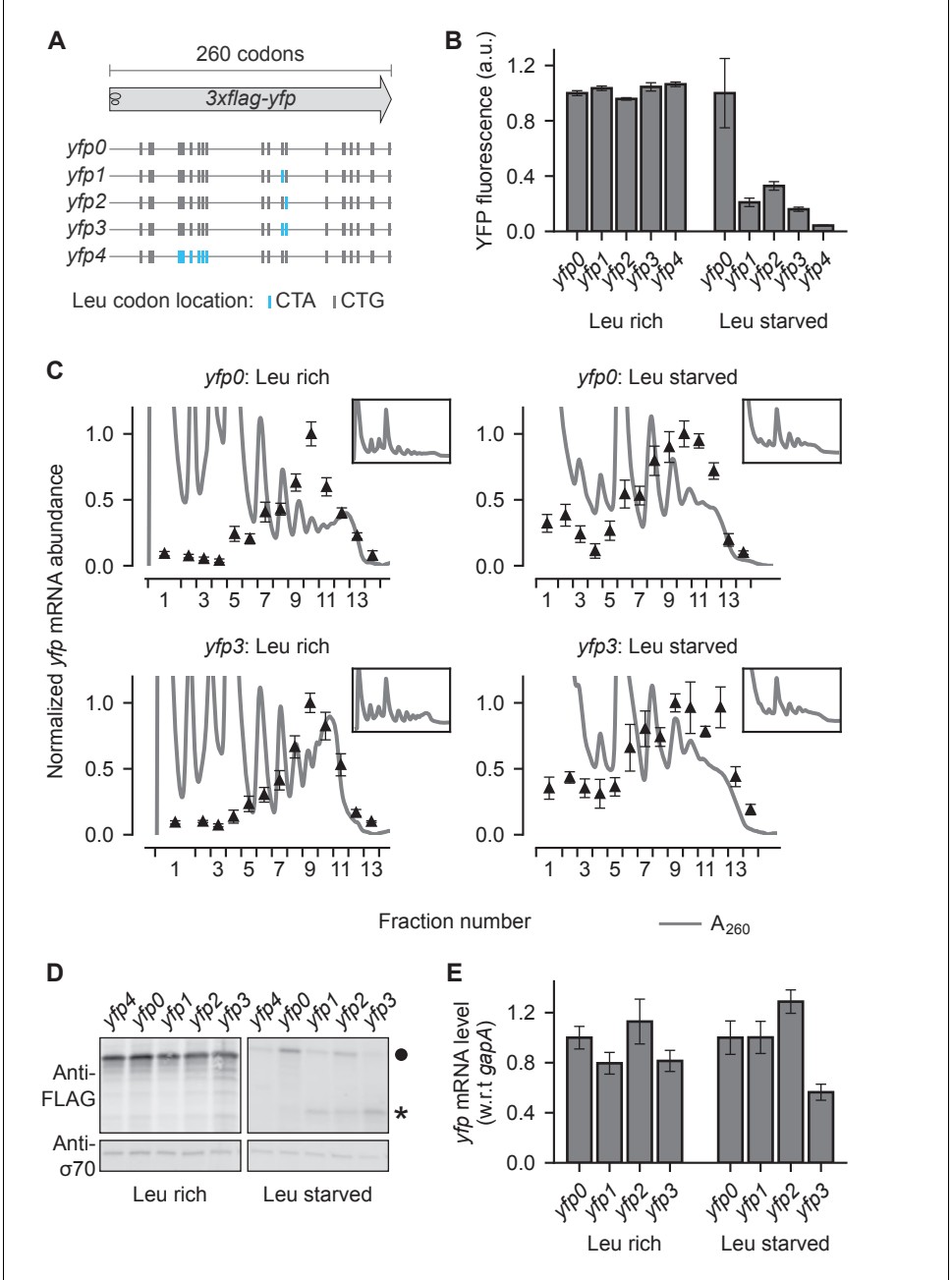

**Figure 1.** Effect of ribosome stalling on measured protein level, mRNA level, and polysome occupancy. (**A**) Schematic of ribosome stalling reporters used in B–E. Blue vertical lines show the location of CTA Leu codons that cause ribosome stalling during Leu starvation in *E coli*. Locations of CTG Leu codons that do not induce ribosome stalling are shown in grey. For experiments in B–E, reporters were induced either in Leu-rich growth medium for 20 min or Leu-starvation medium for 60 min. Schematic of a ribosome with 30 nt footprint is shown for reference. (**B**) YFP fluorescence normalized by that of *yfp0* in each condition. Error bars represent standard error over triplicate cultures. (**C**) Triangles indicate *yfp* mRNA levels in polysomes fractionated through a sucrose gradient, and measured by quantitative RT-PCR spanning the CTA codons. Absorbance at 260 nm ($A_{260}$) is shown in grey with magnified Y axis to highlight polysome profiles. X axis ticks delineate polysome fractions used for the corresponding qPCR measurement. An *in vitro* transcribed luciferase mRNA was spiked in for normalizing the mRNA levels by each fraction's volume. Error bars represent standard error of qPCR over triplicates. *Inset*: Polysome profiles showing full monosome peak. (**D**) *Top panel*: Western blot against the 3xFLAG epitope at the N-terminus of YFP reporter. Circle indicates the size of the full length 3xFLAG-YFP product. Star indicates the size of truncated product expected from abortive termination of ribosomes at CTA codons in *yfp1*–*yfp3*. *Bottom panel*:

*Figure 1 continued*

Western blot against the RNA polymerase σ70 subunit shown as a loading control. (E) *yfp* mRNA levels normalized by that of *yfp0* in each condition. An endogenous mRNA, *gapA*, was used for internal normalization. Error bars represent standard error of qPCR over triplicates.

to background level as observed in our earlier work (*Subramaniam et al., 2013*). Thus, YFP expression can serve as a quantitative readout of the effect of ribosome stalling on protein synthesis.

We then sought biochemical evidence supporting a role for either ribosome traffic jams or abortive termination in the reduction of YFP expression caused by stall-inducing CTA codons. We reasoned that ribosome traffic jams that reduce protein expression by blocking initiation should increase the number of ribosomes on an mRNA when the stall site is far from the initiation region. However, polysome fractionation of leucine-starved *E. coli* did not indicate an unambiguous shift of the *yfp* mRNA to higher polysome fractions when two stall-inducing CTA codons were introduced 475nt from the start codon (*Figure 1C*, top vs bottom panels). This observation agrees with previous ribosome density measurements that detected traffic jams of only 1–2 ribosomes behind stalled ribosomes (*Subramaniam et al., 2014*).

We detected truncated YFP fragments consistent with abortive termination at stall-inducing CTA codons during leucine starvation (*Figure 1D*). Previous studies suggested that abortive termination of stalled ribosomes requires cleavage of mRNA near the stall site as an obligatory step (*Keiler, 2015*; *Hayes and Sauer, 2003*; *Ivanova et al., 2004*). Therefore, we tested whether changes in mRNA levels could account for the 3–4 fold decrease in YFP expression caused by single CTA codons during leucine starvation (*Figure 1B*). However, we found that *yfp* mRNA levels, as measured by quantitative RT-PCR spanning the single CTA codons, did not decrease significantly during leucine starvation (*Figure 1E*). Similarly, introducing two CTA codons resulted in <2 fold decrease in *yfp* mRNA levels despite ~6 fold decrease in YFP expression (*Figure 1E vs 1B*). These observations are consistent with earlier measurements using ribosome profiling and Northern blotting that did not find evidence for significant mRNA cleavage or decay upon ribosome stalling at CTA codons during leucine starvation (*Subramaniam et al., 2014*).

## Computational modeling of ribosome kinetics at stall sites

Since the above reporter-based experiments were qualitative and could miss subtle effects, we formulated an alternate approach using computational modeling to quantitatively test the role of ribosome traffic jams and abortive termination at stall sites during amino acid starvation in *E. coli*. To this end, we defined a minimal set of five kinetic states at ribosome stall sites and the rate constants for transition between these kinetic states (*Figure 2A*).

In our modeling (*Figure 2A*), ribosomes stalled during amino acid starvation are represented by the A-site empty state *ae*. Once the aminoacyl-tRNA is accommodated, A-site empty ribosomes transition to the A-site occupied state *ao*. Ribosomes transition back from the A-site occupied state *ao* to the A-site empty state *ae* upon peptide-bond formation and translocation. Beyond the *ae* and *ao* states, we did not consider additional kinetic states in the normal elongation cycle of the ribosome (*Wintermeyer et al., 2004*; *Blanchard et al., 2004*), since these states cannot be resolved using our measurements of protein synthesis rates during amino acid starvation. Ribosomes that have dissociated from mRNA, due to either normal termination at stop codons or abortive termination at stall sites, transition to the free state *f*. Finally, collision between a stalled ribosome with an empty A-site and a trailing ribosome with an occupied A-site transitions the stalled ribosome to the 5' hit state *5h* and the trailing ribosome to the 3' hit state *3h*.

To model ribosome traffic jams, we chose the rate constant for abortive termination of all elongating ribosomes to be zero. Hence if the duration of ribosome stalling is sufficiently long, a queue of trailing ribosomes forms behind the stalled ribosome and ultimately reduces protein synthesis rate by blocking the initiation region. We designate this as the traffic jam (TJ) model (*Figure 2B*, top).

To model abortive termination, we set the transition rate constant from stalled ribosomes to free ribosomes to be non-zero. Abortive termination occurs selectively at stalled ribosomes, and not at normally elongating ribosomes (*Roche and Sauer, 1999*). Even though the mechanistic basis for this

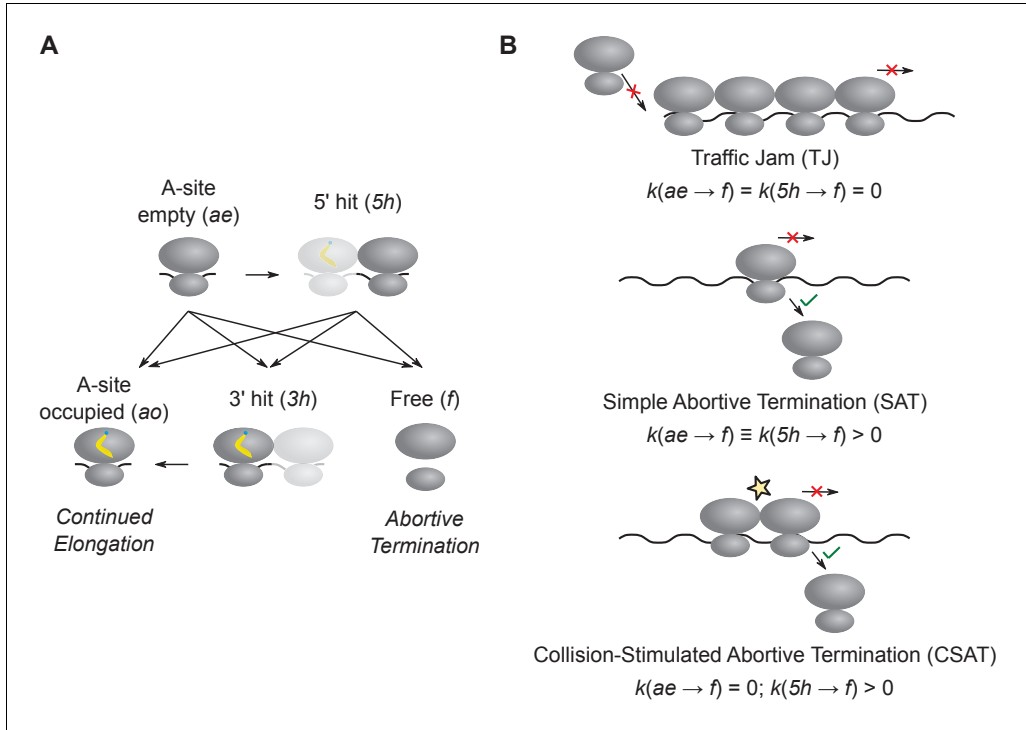

**Figure 2.** Computational modeling of ribosome kinetics at stall sites. (**A**) Distinct ribosome states that were considered during computational modeling of each elongation cycle. (**B**) Schematic of the three kinetic models of stalled ribosomes considered in this work. The three equations indicate the rate of abortive termination from the A-site empty (*ae*) and the 5' hit (*5h*) states in the three different kinetic models. The rate of abortive termination from the A-site occupied (*ao*) and 3' hit (*3h*) states is zero in all three models.

selectivity is poorly understood (*Janssen et al., 2013*; *Miller and Buskirk, 2014*), we can account for the selectivity in our modeling by simply setting the abortive termination rate to be zero at all codons except at the stall site (*Subramaniam et al., 2014*). We designate this as the simple abortive termination (SAT) model (*Figure 2B*, middle).

While abortive termination and traffic jams are usually considered as independent molecular processes (*Kurland, 1992*; *Andersson and Kurland, 1990*), our definition of kinetic states (*Figure 2A*) suggests a more general model in which these processes are coupled. Specifically, we considered a model in which the rate of abortive termination is non-zero only when stalled ribosomes have undergone a collision with a trailing ribosome, i.e. when they are in the *5h* state. We designate this as the collision-stimulated abortive termination (CSAT) model (*Figure 2B*, bottom). As shown below, the CSAT model is closer to experimental measurements of protein synthesis rate than the TJ and SAT models, and it also suggests a mechanistic basis for the selectivity of abortive termination.

## Experimental variables for distinguishing kinetic models of ribosome stalling

Predicting the effect of ribosome stalling on YFP expression in our three kinetic models (*Figure 2B*) requires knowledge of the elongation rate and the abortive termination rate of ribosomes at stall-inducing codons during amino acid starvation in *E. coli*. In principle, these rate constants can be estimated using the ribosome profiling method (*Subramaniam et al., 2014*; *Shah et al., 2013*), but sequence-specific and protocol-related biases in ribosome profiling (*Woolstenhulme et al., 2015*; *Mohammad et al., 2016*; *Lareau et al., 2014*) will introduce a large uncertainty in this estimation. Therefore, we sought to identify experimental variables that would enable us to discriminate between the different kinetic models of ribosome stalling without precise knowledge of the underlying rate constants.

First, we examined the effect of varying the initiation rate of an mRNA with a single stall site in our three kinetic models (*Figure 3A*). We used stochastic simulations to predict the protein synthesis rate from a *yfp* mRNA under this perturbation (Materials and methods). We chose the elongation and abortive termination rate constants at the stall site so that an mRNA with an initiation rate of 0.3 s$^{-1}$ — a typical value for *E. coli* mRNAs (*Kennell and Riezman, 1977*; *Subramaniam et al., 2014*) — had the same protein synthesis rate (number of full proteins produced per unit time) in all three models. In the SAT model, varying the initiation rate does not modulate the effect of the stall site on protein synthesis rate (*Figure 3A*, blue squares). By contrast, in the TJ and CSAT models, the effect of the stall site on protein synthesis rate is reduced at lower initiation rates (*Figure 3A*, green circles and red diamonds). This reduction is more pronounced in the TJ model because, at low initiation rates, ribosome queues do not block the initiation region in the TJ model, while they still lead to collision-stimulated abortive termination in the CSAT model.

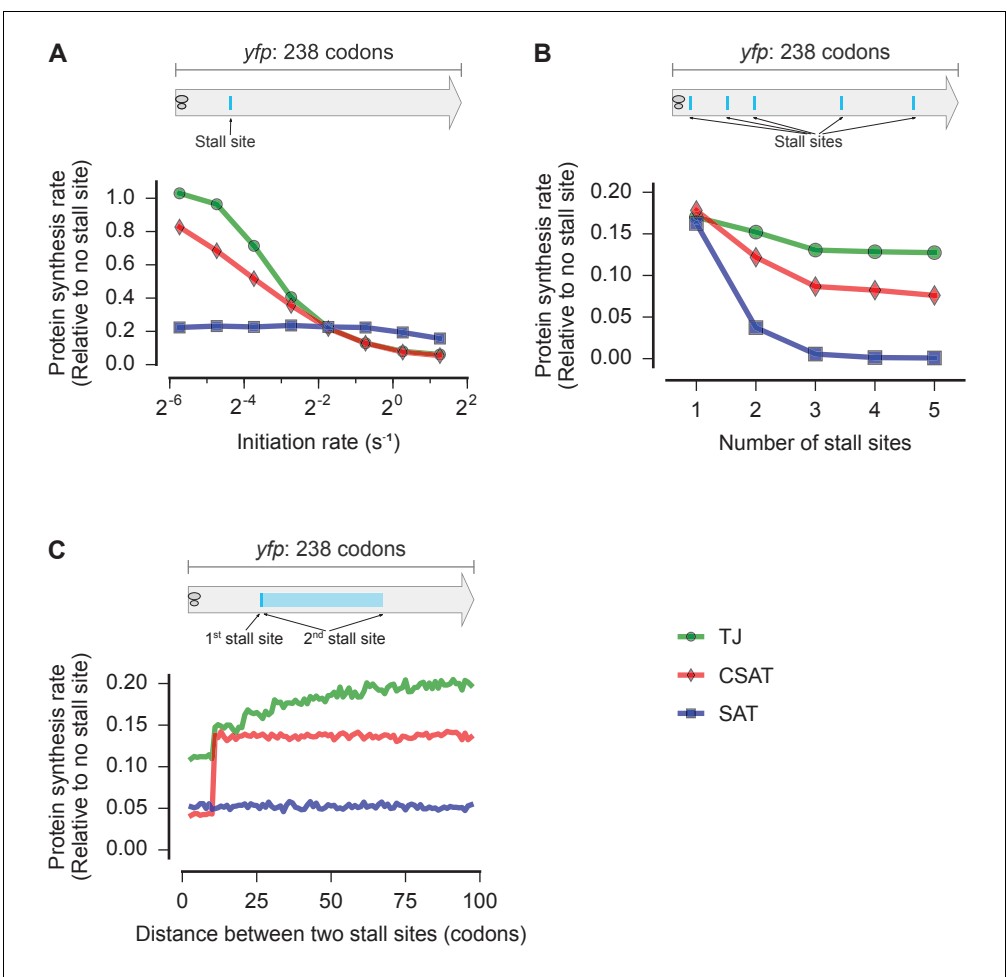

**Figure 3.** Distinct predictions from three kinetic models. Predicted effect on protein synthesis rate upon varying the initiation rate (**A**), the number of ribosome stall sites (**B**), and the distance between two stall sites (**C**) of a *yfp* mRNA. The schematics indicate the locations of the stall sites along *yfp* in our simulation. In B, the stall sites were incrementally added from 5′ to 3′ of the mRNA. In C, the second stall site was located 1 to 100 codons away from the first stall site. Protein synthesis rate for each mRNA is defined as the number of YFP molecules produced per unit time and is shown relative to a *yfp* mRNA without stall sites. The duration of stalling in each model was chosen so that the decrease in protein synthesis rate caused by a single stall site was equal in the three models when the initiation rate is 0.3 s$^{-1}$. $k(ae{\rightarrow}f){\equiv}k(5h{\rightarrow}f)$=1 s$^{-1}$ in the SAT model, and $k(ae{\rightarrow}f)$=0, $k(5h{\rightarrow}f)$=1 s$^{-1}$ in the CSAT model. Other simulation parameters are in *Supplementary file 1*.

Second, we examined the effect of systematically varying the number of stall sites on an mRNA in our three kinetic models (*Figure 3B*). We chose the elongation rate and abortive termination rate constants at stall sites so that the effect of a single stall site on protein synthesis rate was identical between the three models (*Supplementary file 1*). With no further parameter adjustments, we introduced additional identical stall sites, with each stall site separated by at least two ribosome footprints (>60 nt) from other stall sites. In the traffic jam (TJ) model, additional stall sites had very little effect on protein synthesis rate (*Figure 3B*, green circles). In the simple abortive termination (SAT) model, protein synthesis rate decreased exponentially with the number of stall sites (*Figure 3B*, blue squares). In the collision-stimulated abortive termination (CSAT) model, the effect of additional stall sites was intermediate between the TJ and SAT models (*Figure 3B*, red diamonds). The differential effect of multiple stall sites in the three models can be intuitively understood as follows: In the TJ model, extended queues of ribosomes occur only at the first stall site because the average rate at which ribosomes arrive at subsequent stall sites is limited by the rate at which they elongate past the first stall site. In the CSAT model, ribosome collisions occur at a greater rate at the first stall site, but are not completely prevented at subsequent stall sites due to stochastic ribosome elongation past the first stall site. In the SAT model, abortive termination rate at each stall site does not depend on the presence of other stall sites on the mRNA.

Finally, we considered the effect of varying the distance between two identical stall sites in our kinetic models. In the SAT model, varying the distance between two stall sites does not modulate the effect of the stall sites on protein synthesis rate (*Figure 3C*, blue). In the CSAT model, when the two stall sites are separated by less than a ribosome footprint, then the frequency of collisions at the stall sites increases, thus resulting in the lower protein synthesis rate in this regime (*Figure 3C*, red). In the TJ model, the length of ribosome queues at the first stall site is modulated by the formation of shorter ribosome queues at the second stall site when it is within a few ribosome footprints. This interaction results in a lower protein synthesis rate when the stall sites are separated by a few ribosome footprints (*Figure 3C*, green).

## Measured protein synthesis rates support a collision-stimulated abortive termination model

We tested the predictions from our kinetic models using *yfp* reporters with stall-inducing codons during starvation for single amino acids in *E. coli*. First, we measured the effect of varying the initiation rate on the synthesis rate of YFP either by mutating the ATG start codon to a near-cognate codon, or by mutating the Shine-Dalgarno sequence (*Figure 4*, inset). We fitted the ribosome elongation rate at stall-inducing codons in the three kinetic models using the measured YFP synthesis rate for the *yfp* variant with the non-mutated initiation region (variant four in *Figure 4*), and used this fit to predict the YFP synthesis rate of the other initiation mutants with no remaining free parameters (Materials and methods, *Supplementary file 2*). The effect of a single CTA codon on YFP synthesis rate decreased as the initiation rate of the *yfp* variants was reduced (*Figure 4*, black triangles). Both the TJ and CSAT models predicted the decreasing effect of the CTA codon with lower initiation rate (*Figure 4*, green circles and red diamonds). By contrast, the predicted YFP synthesis rate from the SAT model was independent of initiation rate (*Figure 4*, blue squares). This difference between the SAT model, and the TJ and CSAT models was also observed upon introducing CTA, CTC, or CTT codons at other locations in *yfp*, as well as the stall-inducing codon TCG during serine starvation (*Figure 4—figure supplement 1*).

Second, we tested the effect of multiple stall sites on YFP synthesis rate (*Figure 5*). We introduced a single CTA codon at one of five locations among the twenty-two leucine codons in *yfp* (*Figure 5*, inset), and we then combined the single mutations to generate ten *yfp* variants with two CTA codons, two *yfp* variants with three CTA codons, and one *yfp* variant with four CTA codons. We then used the measured YFP synthesis rates (*Figure 5*, black triangles) of the five single CTA variants to fit the ribosome elongation rate at each of the five CTA codon locations in our three kinetic models (Materials and methods, *Supplementary file 3*). These fits, with no remaining free parameters, were used to predict YFP synthesis rates of the multiple-CTA variants during leucine starvation. We found that the TJ model systematically overestimated the YFP synthesis rate for 12 of 13 multiple-CTA variants (*Figure 5*, green circles), while the SAT model systematically underestimated the YFP synthesis rate for all 13 multiple-CTA variants during leucine starvation (*Figure 5*, blue squares). By contrast, the predicted YFP synthesis rates from the CSAT model (*Figure 5*, red diamonds) were

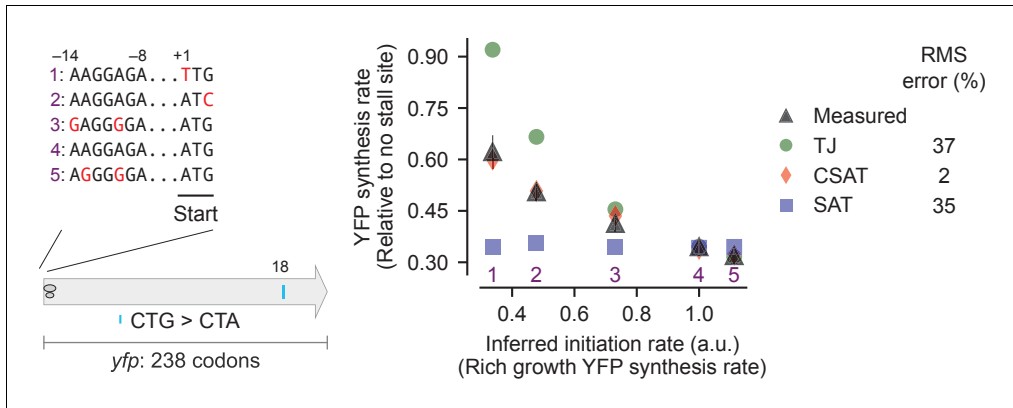

**Figure 4.** Predicted vs. measured YFP synthesis rates during Leu starvation upon variation in initiation rate. *yfp* reporters with a single CTA codon and one of five initiation regions are shown in the schematic. X axis – Measured YFP synthesis rate during Leu-rich growth was used as a proxy for the translation initiation rate. Y axis – Predicted and measured YFP synthesis rates during Leu starvation are shown relative to a *yfp* mRNA without CTA codon. The Leu position 18 is labeled by its order of occurrence along *yfp* relative to the start codon (22 Leu codons total), and corresponds to the 201st codon in *yfp*. Error bars indicate standard error over triplicate cultures. Simulation parameters are shown in ***Supplementary file 2***. RMS error % is the root mean square error between predictions from each model and measured YFP synthesis rate, normalized by the average measured value. RMS error was calculated for initiation region mutants 1, 2, 3 and 5.

The following figure supplement is available for figure 4:

**Figure supplement 1.** Predicted and measured YFP synthesis rates during Leu starvation (A-G) or Ser starvation (H) from *yfp* reporters with a single stall-inducing codon and one of five initiation region variants in ***Figure 4*** (labeled 1–5).

closest to the measured YFP synthesis rates with approximately half the average error of the TJ and SAT models. Similarly, the CSAT model prediction was more accurate when we introduced CTC, CTT, or TCG stall-inducing codons into *yfp* (***Figure 5—figure supplement 1***).

Third, we measured the effect of varying the distance between two stall sites on YFP synthesis rate (***Figure 6***, black triangles). We made pairwise combinations of seven CTA mutations to generate eight variants with a range of distances *d* between the two CTA codons (***Figure 6***, inset). As before, we fitted our three models to the measured YFP synthesis rate of the single CTA variants and used these fits to predict the YFP synthesis rate of the double CTA variants (Materials and methods, ***Supplementary file 4***). We found that two CTA codons separated by less than a ribosome footprint (*d* < 10 codons) resulted in lower protein synthesis rate than two CTA codons separated by several ribosome footprints (*d* > 50 codons) (***Figure 6***, black triangles). This observation was in line with the predictions from the TJ and CSAT models (***Figure 3C***), with the CSAT model providing a better fit than either the TJ or SAT models overall. Similarly, the CSAT model was more accurate when we varied the distance between two CTC codons in *yfp* (***Figure 6—figure supplement 1***).

Combining the results from all the *yfp* mutants (N = 94) and assuming independent and normal distribution of residual errors, we conclude that the TJ model systematically overestimates the measured YFP synthesis rate ($p < 10^{-15}$, one-sided Student's *t*-test), while the SAT model systematically underestimates the measured YFP synthesis rate ($p < 10^{-8}$, one-sided Student's *t*-test). The CSAT model shows no such bias ($p > 0.05$, two-sided Student's *t*-test). Under the same assumption of normal distribution of residual errors and using the Akaike Information Criterion (***Burnham and Anderson, 2013***), we find an Akaike weight >0.999 in favor of the CSAT model over the TJ and SAT models. Thus we conclude that the CSAT model provides a better fit to the measured YFP synthesis rates from the *yfp* mutants than either the TJ or the SAT models when the initiation rate, the number of stall sites, and the distance between stall sites are systematically varied during starvation for single amino acids in *E. coli*.

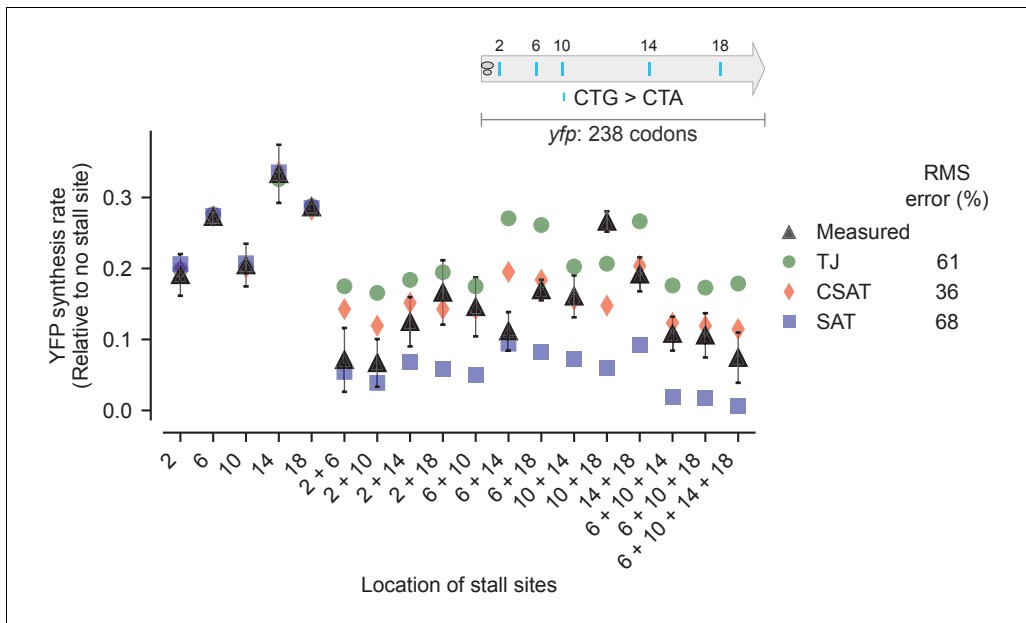

**Figure 5.** Predicted vs. measured YFP synthesis rates during Leu starvation upon variation in number of stall sites. *yfp* reporters having one to four CTA codons at the five Leu positions in *yfp* are shown in the schematic. X axis – location of CTA codons in each of the *yfp* variants. Y axis – Predicted and measured YFP synthesis rates during Leu starvation are shown relative to a *yfp* mRNA without CTA codon. The Leu positions are labeled by their order of occurrence along *yfp* relative to the start codon (22 Leu codons total), and correspond to the following codon positions along *yfp*: 2: 15, 6: 46, 10: 68, 14: 141, 18: 201. Error bars indicate standard error over triplicate cultures. Simulation parameters are shown in **Supplementary file 3**. RMS error % is the root mean square error between predictions from each model and measured YFP synthesis rate, normalized by the average measured value. RMS error was calculated only for mutants with multiple CTA codons.

The following figure supplement is available for figure 5:

**Figure supplement 1.** Predicted and measured YFP synthesis rates during Leu starvation (A-B) or Ser starvation (C) from *yfp* reporters having stall-inducing codons CTC (A), CTT (B), or TCG (C) at one or two among either five Leu positions shown in the upper schematic or six Ser positions in *yfp* in the lower schematic.

## Selectivity, robustness, and ribosome density in the collision-stimulated abortive termination model

The ability of the CSAT model to account for measured YFP synthesis rates from our reporters led us to examine whether this model is consistent with other expected features of ribosome stalling during amino acid starvation in *E. coli*. Specifically, we used our simulations to examine how varying the abortive termination rate affects protein synthesis from mRNAs with and without stall sites, as well as the predicted ribosome density near stall sites in the three kinetic models.

First, only a small fraction of ribosomes are expected to prematurely terminate from mRNAs without stall sites (*Subramaniam et al., 2014*; *Zhang et al., 2010*; *Sin et al., 2016*). Consistent with this expectation, predicted protein synthesis rates from reporters without stall sites did not decrease when the abortive termination rate was increased in the CSAT model (*Figure 7A*, top panel, red). This selectivity towards stalled ribosomes naturally arises in the CSAT model from the requirement for ribosome collisions to cause abortive termination. By contrast, protein synthesis rates from reporters without stall sites decreased with increasing abortive termination rate in a SAT model in which abortive termination was not explicitly specified to be selective for stalled ribosomes (*Figure 7A*, top panel, blue vs. pink).

Second, the frequency of abortive termination is known to be robust to over-expression of factors that rescue stalled ribosomes (*Moore and Sauer, 2005*). Consistent with this observation, we found that increasing the abortive termination rate in the CSAT model predicted only a minor effect on

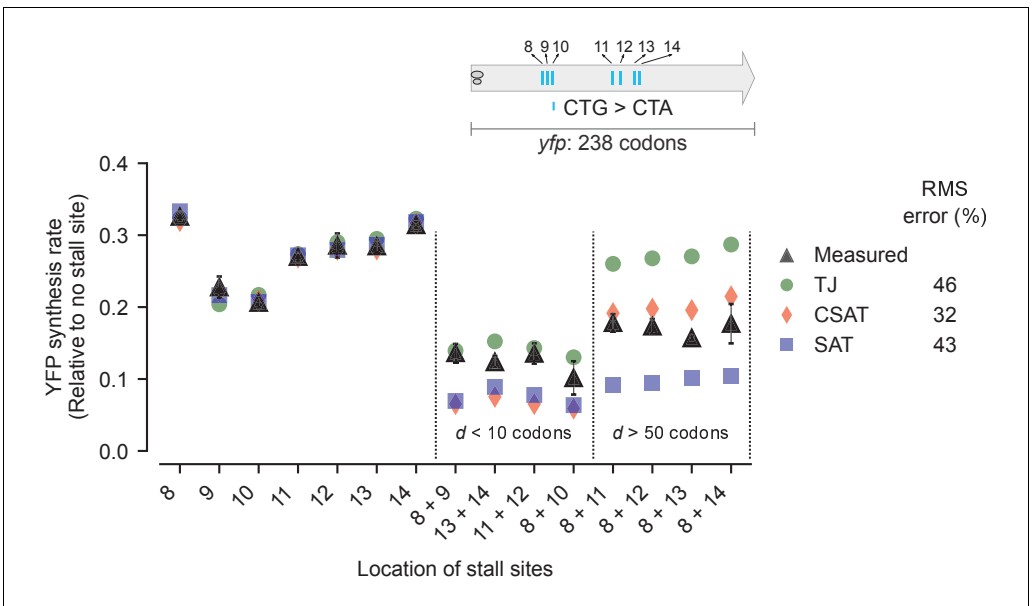

**Figure 6.** Predicted vs. measured YFP synthesis rates during Leu starvation upon variation in distance between stall sites. *yfp* variants with two CTA codons are arranged by increasing value of d, the distance between the CTA codons. X axis – location of CTA codons in each of the *yfp* variants. Y axis – Predicted and measured YFP synthesis rates during Leu starvation are shown relative to a yfp mRNA without CTA codon. The Leu positions are labeled by their order of occurrence along *yfp* relative to the start codon (22 Leu codons total), and correspond to the following codon positions along *yfp*: 8: 60, 9: 64, 10: 68, 11: 119, 12: 125, 13: 137, 14: 141. Error bars indicate standard error over triplicate cultures. Simulation parameters are shown in ***Supplementary file 4***. RMS error % is the root mean square error between predictions from each model and measured YFP synthesis rate, normalized by the average measured value. RMS error was calculated only for mutants with two CTA codons.

The following figure supplement is available for figure 6:

**Figure supplement 1.** Predicted and measured YFP synthesis rates during Leu starvation from *yfp* reporters having CTC codons at one or two among five Leu positions in *yfp* shown in the schematic.

protein synthesis rate from an mRNA with a single stall site (***Figure 7A***, bottom panel, red). By contrast, in both the selective and non-selective SAT models, protein synthesis rate from an mRNA with a single stall site continuously decreased as the abortive termination rate was increased (***Figure 7A***, bottom panel, blue and pink). The robustness of the CSAT model to varying abortive termination rates arises because the frequency of ribosome collisions limit the actual rate of abortive termination at stall sites.

Finally, previous ribosome profiling measurements have detected a queue of only a few ribosomes at CTA codons during leucine starvation in *E. coli* (***Subramaniam et al., 2014***). Consistent with this observation, the length of ribosome queues at the stall site predicted by the CSAT model is limited to a few ribosomes even when the stall site is located ~200 codons from the start codon (***Figure 7B***, red). A similar queue of few ribosomes is also observed in the SAT model (***Figure 7B***, blue and pink). By contrast, the TJ model predicts a queue of ~20 ribosomes when stall sites are located ~200 codons from the start codon (***Figure 7B***, green).

## Discussion

In this work, we used a combination of computational modeling and reporter-based measurements of protein synthesis rate to constrain ribosome kinetics at stall sites during single amino acid starvation in *E. coli*. Our approach allowed us to test two previously proposed models for how ribosome stalling decreases protein expression, namely, ribosome traffic jams that block initiation (TJ model) and simple abortive termination of stalled ribosomes (SAT model). We also considered a novel

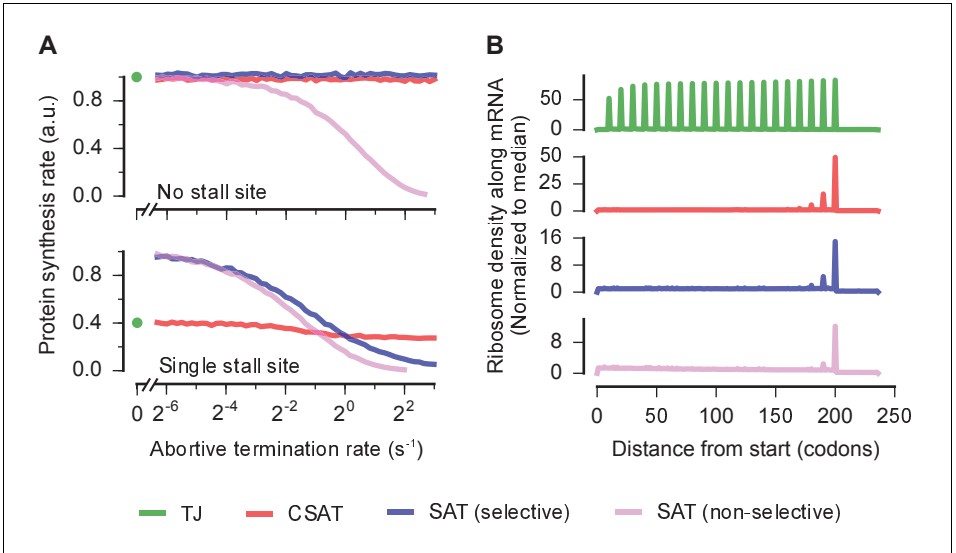

**Figure 7.** Selectivity, robustness, and ribosome density in the collision-stimulated abortive termination model. (**A**) Predicted effect of varying the abortive termination rate on protein synthesis rate from a *yfp* mRNA having no stall site (top panel) or a single stall site (bottom panel). The TJ model corresponds to an abortive termination rate of zero, and is shown as a single point at the left. The selective SAT model has non-zero abortive termination rate at only the codon corresponding to the stall site. The non-selective SAT model has non-zero abortive termination rate at all codons along the *yfp* mRNA. Overlapping curves for CSAT and SAT (selective) models in the top panel were manually offset for clarity. (**B**) Predicted ribosome occupancy on a *yfp* mRNA with a single stall site at the 201st codon. Ribosome occupancy is normalized by its median value across the mRNA. Simulation parameters are shown in *Supplementary file 5*.

model in which ribosome collisions stimulate abortive termination of stalled ribosomes (CSAT model). Our integrated approach allowed us to infer the extent to which each of these three kinetic models quantitatively accounted for the measured protein synthesis rate from a library of *yfp* variants during starvation for single amino acids.

The TJ model has been considered theoretically in several studies (*MacDonald et al., 1968*; *Mitarai et al., 2008*; *Zhang et al., 1994*). While queues of ~7 ribosomes have been detected *in vitro* (*Wolin and Walter, 1988*), ribosome profiling studies have revealed a queue of only a few ribosomes at stall sites *in vivo* (*Woolstenhulme et al., 2015*; *Subramaniam et al., 2014*; *Guydosh and Green, 2014*). These smaller queues can modulate protein expression only if the stall site is within a few ribosome footprints from the start codon (*Mitarai et al., 2008*; *Liljenström and von Heijne, 1987*; *Tuller et al., 2010*). Nevertheless, recent studies on EF-P dependent pauses in bacteria and rare-codon dependent pauses in yeast suggested that the TJ model underlies the decreased protein expression when stall sites are over 100 codons away from the start codon (*Hersch et al., 2014*; *Chu et al., 2014*). These conclusions were based on observations that decreasing initiation rate of ribosomes on reporters reduced the effect of stall sites on protein expression (*Hersch et al., 2014*; *Chu et al., 2014*). This regulatory effect of initiation rate was also observed in our experiments (*Figure 3*). However, we find that both the TJ and CSAT models predict this regulatory effect of initiation rate (*Figure 3A*), while only the CSAT model predicts a queue of few ribosomes (*Figure 7B*) that is observed experimentally. Thus, collision-stimulated abortive termination is a plausible alternative mechanism to the traffic jam model proposed in previous studies (*Hersch et al., 2014*; *Chu et al., 2014*).

Simple kinetic partitioning between normal elongation and abortive termination has been proposed as a possible mechanism for how ribosome rescue factors might act at ribosomes that are stalled within an mRNA (*Brandman and Hegde, 2016*; *Shao and Hegde, 2016*). However, our modeling indicates that this non-selective mechanism of abortive termination will result in decreased protein expression from mRNAs that do not have stall sites (*Figure 7A*, top panel). This observation

can be intuitively understood from the fact that even a small probability of abortive termination during each elongation cycle will be exponentially amplified over the course of translating a typical *E. coli* protein with 300 amino acid residues.

Despite the better fit provided by the CSAT model to our measured YFP synthesis rates, there still remains a residual error in its prediction (*Figures 4*, *5* and *6*). This error might arise from several simplifying assumptions in our definition of the CSAT model, which we made in order to emphasize its qualitative difference with the TJ and SAT models (*Figure 2B*). First, we assumed the rate of abortive termination to be zero in the absence of ribosome collisions. Relaxing this assumption is likely to provide a better fit to our measurements, but it will introduce an extra free parameter while not providing additional mechanistic insight into the kinetics of abortive termination. Second, we assumed the rate of abortive termination to be zero for the trailing ribosome in the collided state (*3h* in *Figure 2B*), since there is no biochemical evidence for such a process. This assumption could be relaxed based on evidence from future biochemical studies of ribosome queues formed at stall sites. The inverse approach used in our work relied on model predictions that did not depend sensitively on underlying kinetic parameters such as the elongation rate and the abortive termination rate at stall sites. Hence, our work cannot be used to infer the exact values of these kinetic parameters *in vivo*. Finally, we studied the CSAT model solely in the context of ribosome stalls caused by amino acid starvation in *E. coli*. Hence, the validity of this model at ribosome stalls in exponentially growing bacterial cells remains to be tested.

Ribosome collisions during amino acid starvation could stimulate abortive termination through several mechanisms. Specifically, ribosome collisions could either stimulate spontaneous drop-off of stalled ribosomes, or they could stimulate the activity of quality control pathways such as the tmRNA and the ArfA systems that rescue stalled ribosomes (*Keiler, 2015*; *Shoemaker and Green, 2012*). In the latter case, ribosome collisions might allow the quality control pathway to selectively recognize ribosomes that have been stalled for an extended duration over ribosomes that are transiently stalled due to the stochasticity of normal elongation (*Miller and Buskirk, 2014*). In this sense, the frequency of ribosome collisions can provide a natural timer for achieving selectivity of quality control pathways towards stalled ribosomes (*Shao and Hegde, 2016*). Further, the robustness of the CSAT model to changes in the abortive termination rate (*Figure 7A*, bottom panel) can buffer against cell-to-cell variation in the concentration of quality control factors that mediate abortive termination. Finally, ribosome collisions might also have a role in stimulating the activity of eukaryotic translational quality control pathways (C. Simms and H. Zaher, personal communication) such as No-Go mRNA decay (*Doma and Parker, 2006*), where the kinetic events leading to recognition of stalled ribosomes remain poorly defined (*Shoemaker and Green, 2012*). This general role for ribosome collisions in translational quality control could have arisen during evolution to minimize the idling of translation-competent ribosomes on mRNAs.

## Materials and methods

### Bacterial strains and plasmids

All leucine starvation experiments in this study were performed using an *E. coli* strain (*Subramaniam et al., 2013*) that is auxotrophic for leucine and contains the *tet* repressor gene for inducible control of reporter genes (ecMF1). Serine starvation experiments were performed using a similar strain, but auxotrophic for serine instead of leucine (ecMF403). All fluorescent reporters in this study were cloned into a very low copy expression vector (SC*101 *ori*, 3–4 copies per cell) used in our previous work (*Subramaniam et al., 2013*) (pASEC1, Addgene plasmid #53241). The fluorescent reporter genes used in leucine starvation experiments were based off a yellow fluorescent protein sequence (*yfp0*) present in pASEC1, which encodes a fast-maturing 'Venus' variant of YFP. All 22 leucine codons in *yfp0* were chosen as CTG. All *yfp* reporters used for serine starvation experiments were constructed from a *yfp* variant that had the AGC codon at all eight serine positions. For constructing *yfp* reporters with single stall sites during leucine starvation, the corresponding CTG codon in *yfp0* was mutated to CTA, CTC, or CTT by encoding these mutations in oligos and using Gibson assembly (*Gibson et al., 2009*). The single stall reporters for serine starvation were similarly constructed by mutating a single AGC codon to TCG codon. A *yfp* variant with seven leucine codons mutated to CTA was used in all plate reader experiments as a control for the lower limit of detection

of YFP fluorescence under Leu starvation (ecMF112). Variants of *yfp* with multiple CTA, CTC, CTT, or TCG codons were constructed by Gibson assembly of PCR fragments from the corresponding single codon variants of *yfp*. The start codon and Shine-Dalgarno sequence variants of *yfp* were generated by encoding these mutations in one of the PCR oligos for *yfp*. The *3xflag-yfp* variants were generated by the addition of a 22 codon sequence at the 5' end that encoded a 3X-FLAG peptide used in our previous work (*Subramaniam et al., 2013*). All strains and plasmids used in this study are available upon request (See *Supplementary file 7* for list of strains and plasmids).

## Growth and fluorescence measurements

Overnight cultures were inoculated in biological triplicates from freshly grown single colonies or patched colonies from glycerol stocks. Overnight cultures were grown in a modified MOPS rich defined medium (*Subramaniam et al., 2013*; *Neidhardt et al., 1974*) made with the following recipe: 10X MOPS rich buffer, 10X ACGU nucleobase stock, and 100X 0.132M K2HPO4 were used at 1X final concentration as in the original recipe. In addition, the overnight growth medium contained 0.5% glucose as carbon source and 800 µM of 19 amino acids and 10 mM of serine. pH was adjusted to 7.4 using 1M NaOH and appropriate selective antibiotic (100 µg/ml carbenicillin) was added. 200 ng/ml of anhydro-tetracycline (aTc) was also added in order to induce the PLtetO-1 promoter (*Lutz and Bujard, 1997*). 1 ml overnight cultures were grown in 2 ml deep 96-well plates (AB0932, Fisher) at 30°C with shaking at 1200 rpm (Titramax 100 shaker) for 12 to 16 hr.

For amino acid starvation time course experiments, overnight cultures were diluted 1:100 into 150 µl of the same MOPS rich-defined medium as the overnight cultures. However, leucine was added at 100 µM and supplemented with its methyl ester analog at 160 µM (AC125130250, Fisher) for leucine starvation experiments. Similarly, serine was added at 5 mM and supplemented with its methyl ester analog at 800 µM (412201, Sigma) for serine starvation experiments. Addition of each methyl ester results in a steady but limiting supply of the amino acid due to slow hydrolysis of the ester, and this enables extended and accurate measurements of protein synthesis rate under the amino acid starvation condition (*Subramaniam et al., 2013*). Except for the limiting amino acid, the remaining 19 amino acids were present at the overnight culture concentrations during the amino acid starvation experiments.

Diluted overnight cultures were grown in 96-well plates (3595, Costar) at 30°C with shaking at 1200 rpm (Titramax 100 shaker). A 96-well plate reader (Infinite M1000 PRO, Tecan) was used to monitor cell density (absorbance at 600 nm) and YFP synthesis (fluorescence, excitation 504 nm and emission 540 nm). Each plate was read every 15 min and shaken in between readings for a total period of 6–10 hr.

For experiments in *Figure 1*, overnight cultures were grown without aTc and diluted 1:1000 into the same medium. Then when the $OD_{600}$ reached 0.5, the cells were spun down at 3000 g for 5 min and then re-suspended in the same medium, but either with or without leucine, and with aTc for reporter induction. Fluorescence, Western blots, and qRT-PCR measurements in *Figure 1* were performed from these cultures after shaking at 37°C, 200 rpm for 20 min with leucine or 60 min without leucine.

## Polysome profiling

Overnight cultures were diluted 1:200 into 400 ml MOPS rich defined medium and grown at 37°C to an $OD_{600}$ of 0.2. Cells were harvested by vacuum filtration on a 0.2 µm nitrocellulose membrane (BA83, GE) and subsequently cut in half. One half was added to 200 ml MOPS rich defined medium, the other to 200 ml of same medium but without leucine. After growth at 37°C for either 20 min (Leu-rich cultures) or 1 hr (Leu starvation cultures), cells were harvested by vacuum filtration again. Cells were scraped from the membrane using a plastic spatula before the membrane became dry, and then immediately submerged in liquid nitrogen and stored at –80°C. Frozen cells were then re-suspended in 0.7 ml bacterial lysis buffer (20 mM Tris pH 8.0, 10 mM MgCl2, 100 mM NH4Cl, 2 mM DTT, 0.1% NP-40, 0.4% Triton X-100, 100 U/ml DNase I, and 1 mM chloramphenicol) and lysed using glass beads (G1277, Sigma) by vortexing 4 × 30 s at 4°C with 60 s cooling on ice in between. The lysate was clarified by centrifugation at 21,000 g, 4°C for 10 min and supernatant was transferred to a fresh tube.

Lysate RNA concentration was quantified by $A_{260}$ (Thermo Scientific Nanodrop) and 100–200 µl of lysate containing 0.5 mg RNA was loaded onto a 10–50% sucrose gradient made with 20 mM Tris pH 8.0, 10 mM MgCl2, 100 mM NH4Cl, and 2 mM DTT. Polysomes were separated by centrifugation in an SW41 rotor at 35,000 rpm for 3 hr at 4°C. Gradients were then fractionated into 15 fractions containing 25.6 ng spike-in control firefly luciferase mRNA. RNA from each fraction was column-purified along with in-column DNase I digestion (Quick-RNA Miniprep, Zymo Research, Irvine, CA).

## Total RNA extraction

Phenol-chloroform extraction method was used to obtain total RNA. 10 ml of cells were quickly chilled in an ice water bath and harvested by centrifugation at 3000 g for 5 min. Cell pellets were re-suspended in 500 µl of 0.3 M sodium acetate and 10 mM EDTA pH 4.5. Re-suspended cells were mixed with 500 µl of acetate-saturated phenol-chloroform pH 4.5 and 500 µl of acid-washed glass beads (G1277, Sigma). The mixture was shaken in a vortexer for 3 min and then clarified by centrifugation at 21,000 g for 10 min. The samples were maintained at 4°C through this step. The aqueous layer was extracted twice with acetate-saturated phenol-chloroform pH 4.5 and once with chloroform. Total RNA was precipitated with an equal volume of isopropanol, washed with 70% ethanol, and finally re-suspended in 200 µl of RNase-free 10 mM Tris pH 7.0. 200 ng of the total RNA was treated with DNase I (M0303S, NEB) to remove residual DNA contamination (manufacturer's instructions were followed). The DNA-free RNA was column-purified (Quick-RNA Miniprep, Zymo Research, Irvine, CA).

## Reverse transcription and quantitative PCR

Reverse transcription (RT) was performed using 10–20 ng of DNA-free RNA and Maxima reverse transcriptase (EP0741, Thermo), used according to manufacturer's instructions. Random hexamer primers were used for priming the RT reaction. At the end of the RT reaction, the 10 µl RT reaction was diluted 20-fold and 5 µl of this diluted sample was used as template for qPCR in the next step. qPCR was performed using Maxima SYBR Green/ROX qPCR Master Mix (FERK0221, Thermo) and manufacturer's instructions were followed. qPCR was performed in triplicates for each RT reaction and appropriate negative RT controls were used to confirm the absence of DNA contamination. *gapA* mRNA was used as internal reference to normalize all other mRNA levels. Primers for qPCR were from our previous work (*Subramaniam et al., 2013*). *Δ*Ct method was used to obtain relative mRNA levels. Analysis was implemented using Python 2.7 libraries. Code for analysis and plotting of figures starting from raw qPCR data is publicly available at http://github.com/rasilab/ferrin_elife_ 2017 (*Subramaniam, 2017*) as Jupyter notebooks (*Perez and Granger, 2007*).

## Western blotting

Cells were harvested by centrifugation and protein was precipitated by mixing trichloroacetic acid to a final concentration of 10%. The mixture was incubated on ice for 15 min and the supernatant was removed. Protein pellets were re-suspended in 100 µl 1X Laemmli Buffer (Biorad), boiled at 99°C for 5 min, and then loaded onto each lane of a 4–20% polyacrylamide gel (Biorad) and SDS-PAGE was carried out at 200V for 50 min. Proteins were transferred to a nitrocellulose membrane at 500mA for 60 min using a wet-transfer apparatus (Biorad). The membrane was cut along the 50kD marker and both halves were blocked in Odyssey PBS Blocking Buffer (Li-cor) for 60 min. The lower-MW half was incubated with a 1:6000 dilution of an anti-FLAG antibody (F3165, Sigma), and the higher-MW half in the same dilution of an anti-$\sigma_{70}$ antibody (WP004, Neoclone), each in 15 ml of Odyssey PBS Blocking Buffer with shaking at 4°C overnight. After washing 4 × 5 min with TBST, the membrane was incubated with 1:10,000 dilution of a secondary dye-conjugated antibody (925–68072, Li-cor) in 15 ml of Odyssey PBS Blocking Buffer with shaking at room temperature for 60 min. After washing 4 × 5 min with PBS, the membrane was imaged using a laser-based fluorescence imager.

## Growth and fluorescence data analysis

$OD_{600}$ and YFP fluorescence were recorded as time series for each well of a 96-well plate. Background values for $OD_{600}$ and YFP fluorescence were subtracted based on measurements from a well with just growth medium. Time points corresponding to Leu-rich growth and Leu starvation were

identified by manual inspection of $OD_{600}$ curves. The onset time of starvation was automatically identified as the time point at which $YFP/OD_{600}$ reached a minimum value. YFP synthesis rate during Leu-rich exponential growth was defined as the average of $YFP/OD_{600}$ values for the three points around the onset time of starvation. YFP synthesis rate during Leu starvation was defined as the slope of a linear fit to the fluorescence time series in the Leu starvation regime. YFP synthesis rates for individual wells were averaged over biological replicate wells for calculation of mean and standard error. Analysis was implemented using Python 2.7 libraries. Code for analysis and plotting of figures starting from raw plate reader data is publicly available at http://github.com/rasilab/ferrin_elife_2017 (*Subramaniam, 2017*) as Jupyter notebooks (*Perez and Granger, 2007*).

## Simulation

The kinetic models in *Figure 2* were implemented as stochastic simulations in the C++ object-oriented programming language. Separate classes were defined to represent ribosomes, mRNA transcripts, gene sequences, tRNAs, and codons. Each elongating ribosome was represented as an instance of the *Ribosome* class. The four distinct states of the elongating ribosome (*ae, ao, 5h, 3h* in *Figure 2A*) were tracked using three *bool* properties of the *Ribosome* class: *AsiteEmpty*, *hitFrom5-Prime*, and *hitFrom3Prime*. The identities of the tRNAs occupying the A-site and P-site of the elongating ribosome were tracked. Only the aggregate number of ribosomes in the free state (*f* in *Figure 2A*) was tracked. Instances of the *transcript* class were used to track the number of proteins produced from each transcript. The *gene*, *tRNA* and *codon* classes were used as data structures and their properties did not change during the course of the simulation.

Since our reporters were expressed from very low copy number plasmids, translation of the reporter mRNAs is not expected to perturb the native translation machinery in the cell. Therefore we assumed that both the translation rate of native mRNAs, as well as the pool of free ribosomes and aminoacyl-tRNAs remain constant across all reporters used in this study. Hence, each simulation considered a minimal set of two mRNA molecules that both encoded YFP. The first mRNA molecule was a control *yfp* sequence without any CTA, CTC or CTT codon. The second mRNA molecule was the test *yfp* sequence with the CTA, CTC or CTT codon as specified for individual simulations. The simultaneous translation of the two mRNA molecules was simply to ensure that we used exactly the same set of parameters for our test and control reporters during simulation runs and subsequent analyses.

We simulated four different molecular processes during translation: initiation, elongation, aminoacylation and abortive termination. The rates of all other steps in translation such as termination and ribosome recycling were set to be instantaneous.

The initiation rate of all mRNA sequences was set as 0.3 s$^{-1}$ [a typical value for *E. coli* mRNAs (*Kennell and Riezman, 1977*; *Subramaniam et al., 2014*)] except when this rate was explicitly varied, either to demonstrate its effect in our kinetic models (*Figure 3A*) or for experimental fits (*Figure 4*, *Figure 4—figure supplement 1*). For the experimental fits in *Figure 4* and *Figure 4—figure supplement 1*, the measured YFP synthesis rate of the initiation region mutants during Leu-rich growth relative to the starting sequence (four in *Figure 4*) was used to scale the default initiation rate of 0.3 s$^{-1}$.

Elongation cycle of ribosomes at each codon was divided into two steps:

In the first elongation step, the cognate tRNA is accommodated into the A-site. The rate of tRNA accommodation was chosen to be non-zero only when ribosomes are in the *ae* state. The tRNA accommodation rate for all codons was calculated as the product of a pseudo first-order rate constant ($2 \times 10^7$ M$^{-1}$s$^{-1}$), the concentration of individual tRNAs, and a weight factor to account for codon-anticodon pairing strength. The concentration of tRNAs and the weight factors were based on measured concentration of *E. coli* tRNAs (*Dong et al., 1996*) and known wobble-pairing rules (*Subramaniam et al., 2014*; *Shah et al., 2013*). Leucine starvation was simulated using a previous whole cell model of translation (*Subramaniam et al., 2014*). The steady-state charged fraction of all tRNAs from this whole-cell model during leucine starvation was used for our *yfp* reporter simulation as the default values. To fit the measured YFP synthesis rate of single stall-site variants (*Figures 4*, *5* and *6*, *Figure 4–figure supplement 1*, *Figure 5–figure supplement 1*, *Figure 6–figure supplement 1*), the tRNA accommodation rate at CTA, CTC and CTT codons was systematically varied in the three kinetic models. These fit values were used for illustrating the predictions from the kinetic models in *Figure 3* and *Figure 7*.

In the second elongation step, peptide bond is formed and ribosomes translocate to the next codon. This rate was set to be 22 s$^{-1}$ and equal to the maximum measured rate of *in vivo* elongation (*Bremer and Dennis, 1996*).

The aminoacylation rate for all tRNAs was calculated as the product of a pseudo first-order rate constant ($2 \times 10^{10}$ M$^{-1}$s$^{-1}$) and the concentration of individual tRNAs. Even though we simulated this process explicitly, we did not lower this rate for leucine tRNAs to simulate leucine starvation; Instead, we accounted for leucine starvation by using the steady-state charged fraction of leucine tRNAs from our whole-cell model as mentioned above in our discussion of tRNA elongation rate. This modified procedure enabled us to simulate the translation of just the *yfp* reporters without considering all the endogenous mRNAs in the cell.

The abortive termination rate was set to a value of 1 s$^{-1}$ in the SAT and CSAT models and 0 s$^{-1}$ in the TJ model, except when this rate was explicitly varied (*Figure 7A*). We chose this rate to be of the same approximate value as in our ribosome profiling studies (*Subramaniam et al., 2014*). The exact value of this rate is not critical in our SAT and CSAT models since the fitted value of the elongation rate varies accordingly to reproduce the measured protein synthesis rate from our YFP reporters with single stall sites.

The simulations used a stochastic Gillespie algorithm that was implemented in earlier studies (*Subramaniam et al., 2014*; *Shah et al., 2013*). Each simulation was run until 10,000 full-length YFP molecules were produced from the control *yfp* mRNA without stall-inducing codons. The number of full-length YFP molecules produced in the same duration from the second *yfp* mRNA with stall-inducing codons was used to calculate the YFP synthesis rate (in *Figures 3*, *4*, *5*, *6* and *7*, *Figure 4—figure supplement 1*, *Figure 5—figure supplement 1*, *Figure 6—figure supplement 1*) after normalizing by 10,000. Time-averaged ribosome density on each mRNA was also tracked during the simulation run after 100 YFP molecules were produced from the first *yfp* mRNA, and this density was median-normalized for plotting in *Figure 7B*.

Code for creating simulation input files, running the simulation, and plotting of figures starting from simulation results is publicly available at http://github.com/rasilab/ferrin_elife_2017 (*Subramaniam, 2017*) as Jupyter notebooks (*Perez and Granger, 2007*). Parameters common to all simulations are listed in *Supplementary file 6*. Parameters specific to simulations in individual figures are listed in *Supplementary files 1–5*.

## Data accession

Raw data and programming code for reproducing all figures in this paper is publicly available at: http://github.com/rasilab/ferrin_elife_2017 (*Subramaniam, 2017*, with a copy archived at https://github.com/elifesciences-publications/ferrin_elife_2017).

# Acknowledgements

We thank Robert Bradley, Allen Buskirk, Erick Matsen, Premal Shah, Kevin Wood, Hani Zaher, and Brian Zid for discussions. Funding for this work was provided by NIH grant R35 GM119835, NIH grant R00 GM107113, and startup funds from the Fred Hutchinson Cancer Research Center. The computations in this paper were run on the Gizmo cluster supported by the Scientific Computing group at the Fred Hutchinson Cancer Research Center. ARS dedicates this work to his late father, Perinkulam Ramnathan.

# Additional information

### Funding

| Funder | Grant reference number | Author |
| --- | --- | --- |
| National Institute of General Medical Sciences | R35 GM119835 | Michael A Ferrin<br>Arvind R Subramaniam |
| Fred Hutchinson Cancer Research Center | | Michael A Ferrin<br>Arvind R Subramaniam |
| National Institute of General Medical Sciences | R00 GM107113 | Michael A Ferrin<br>Arvind R Subramaniam |

The funders had no role in study design, data collection and interpretation, or the decision to submit the work for publication.

## Author contributions
MAF, Conceptualization, Data curation, Investigation, Writing—original draft, Writing—review and editing; ARS, Conceptualization, Data curation, Software, Formal analysis, Supervision, Investigation, Writing—original draft, Writing—review and editing

## Author ORCIDs
Michael A Ferrin, http://orcid.org/0000-0002-9899-1169
Arvind R Subramaniam, http://orcid.org/0000-0001-6145-4303

## Additional files

### Supplementary files
• Supplementary file 1. Simulation parameters for *Figure 3*

• Supplementary file 2. Simulation parameters for *Figure 3* (including Figure supplement)

• Supplementary file 3. Simulation parameters for *Figure 5* (including Figure supplement)

• Supplementary file 4. Simulation parameters for *Figure 6* (including Figure supplement)

• Supplementary file 5. Simulation parameters for *Figure 7*

• Supplementary file 6. Parameters common to all simulations

• Supplementary file 7. List of strains and plasmids

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
