## [Decision Letter]

Thank you for submitting your article "Kinetic modeling predicts a stimulatory role for ribosome collisions at elongation stall sites in bacteria" for consideration by *eLife*. Your article has been reviewed by three peer reviewers, one of whom, Alan G Hinnebusch (Reviewer #1), is a member of our Board of Reviewing Editors and the evaluation has been overseen by James Manley as the Senior Editor. The following individual involved in review of your submission have agreed to reveal their identity: Yitzhak Pilpel (Reviewer #2).

The reviewers have discussed the reviews with one another and the Reviewing Editor has drafted this decision to help you prepare a revised submission.

Summary:

This study uses a combination of molecular biology and computational modeling to examine the mechanism of reduced translation of mRNA in response to prolonged pausing of elongating ribosomes evoked by particular leucine codons in bacterial cells starved for leucine. It seeks to distinguish between two leading models of simple abortive termination (SAT) versus a traffic jam (TJ) wherein the stalled ribosome sets up a queue of trailing ribosomes that extends to the start codon and inhibits initiation. The latter model has been favored because it can explain the finding that reducing the rate of initiation can mitigate the inhibitory effect of pausing; however, it is at odds with results of ribosome profiling in which long queues of stalled ribosomes have not been observed. The study begins by providing evidence at odds with the TJ model by showing that introducing stall-inducing codons at a large distance (>400 nt) from the initiation codon of a YFP reporter evoked a large reduction in protein output, which is inconsistent with the limited queues of only 1-2 ribosomes (~20 codons) observed previously by ribosome profiling. They also found no significant increase in the average polysome size of the reporter mRNA that would expected from a long queue of trailing ribosomes; and they ruled out the possibility that the mRNA with stalled ribosomes was being degraded.

They then proceed to computational modeling to predict the effects of abortive termination vs. traffic jams on YFP synthesis rates in the presence of different numbers of stall sites, different spacing between two stall sites, and different initiation rates, revealing predicted synthesis rates that vary significantly among three different mechanisms modeled, including TJ (where the stalled and queuing ribosomes never terminate or dissociate from the mRNA; SAT (where only the stalled ribosome terminates and dissociates); and collision-stimulated abortive termination (CSAT, wherein termination of a stalled ribosome and dissociation occurs only on collision with the trailing elongating ribosome. The modeling predicts that increasing the initiation rate has no effect on YFP synthesis in the SAT model (where collisions by trailing ribosomes have no impact), but progressively reduces YFP synthesis in the TJ and CSAT models, where the effects of queuing ribosomes are paramount. The TJ and CSAT models also make predictions quite distinct from those of SAT when the number of stall sites increases, with SAT predicting much steeper and extensive declines, and the TJ and CSAT predictions differing in that additional stall sites have almost no impact for the TJ model. Increasing the distance between two stall sites is predicted to have no impact on YFP synthesis in the SAT model, to evoke a large step increase in synthesis in the CSAT models when the distance between stall sites is less than the ribosome footprint, and in the TJ model, YFP synthesis is predicted to exhibit a similar step increase followed by a gradual further increase that plateaus in the TJ model.

They went on to measure the effects on YFP reporter expression in response to altering the initiation rate with a single stall site, achieved by replacing the ATG start codon with near-cognates, or altering the Shine-Dalgarno sequence. They consistently observed the increased inhibitory effect with increasing initiation rate predicted by the TJ and CSAT models, and at odds with the SAT model, for various positions or sequences of a single stall site. Testing the effect of multiple stall sites in a variety of different configurations showed a better fit of the experimental data for the CSAT versus both the SAT and TJ models. The CSAT model also outperformed the other two models for the constructs where the distance between two stall sites was varied. They finish by presenting modeling results designed to show that the CSAT model out-performs the SAT model in explain in vivo observations of very low rates of premature termination from mRNAs lacking stall sites and the insensitivity of premature termination rates to overexpression of ribosome rescue factors; and the CSAT model outperforms the TJ model in explaining in vivo observations of very limited queuing of ribosomes upstream from stall sites. The authors conclude that collision-induced abortive termination is the most likely mechanism to account for the inhibitory effect of elongation stalling on translational output in *E. coli*.

Essential revisions:

It is necessary to respond to requests for revisions of text and figures to make the paper easier to read and understand. To address the key comment of Rev. #2, it is necessary to repeat a subset of the experiments with a second set of reporters containing a different codon besides leucine to rule out the possibility of effects specific to leucine starvation. One possibility would be to repeat only the analysis in Figure 4 for a different amino acid. To address the key comment of Rev. #3, it is important to acknowledge that the CSAT mechanism might not apply to ribosome stalling imposed by different mechanisms operating in non-amino acid starved cells. Second, it is necessary to perform statistical analyses to determine whether the agreement of the experimental data with the CSAT model predictions is significantly better than that given by the SAT or TJ models. Third, you are asked to repeat the simulations to consider the more realistic situation where the reporter mRNAs with inserted Leu codons are being translated in the presence of a full complement of mRNAs and not merely a single control reporter mRNA lacking the Leu codon.

Reviewer #1:

Summary of work:

This study uses a combination of molecular biology and computational modeling to examine the mechanism of reduced translation of mRNA in response to prolonged pausing of elongating ribosomes evoked by particular leucine codons in bacterial cells starved for leucine. It seeks to distinguish between two leading models of simple abortive termination (SAT) versus a traffic jam (TJ) wherein the stalled ribosome sets up a queue of trailing ribosomes that extends to the start codon and inhibits initiation. The latter model has been favored because it can explain the finding that reducing the rate of initiation can mitigate the inhibitory effect of pausing; however, it is at odds with results of ribosome profiling in which long queues of stalled ribosomes have not been observed. The study begins by providing evidence at odds with the TJ model by showing that introducing stall-inducing codons at a large distance (>400 nt) from the initiation codon of a YFP reporter evoked a large reduction in protein output, which is inconsistent with the limited queues of only 1-2 ribosomes (~20 codons) observed previously by ribosome profiling. They also found no significant increase in the average polysome size of the reporter mRNA that would expected from a long queue of trailing ribosomes; and they ruled out the possibility that the mRNA with stalled ribosomes was being degraded.

They then proceed to computational modeling to predict the effects of abortive termination vs. traffic jams on YFP synthesis rates in the presence of different numbers of stall sites, different spacing between two stall sites, and different initiation rates, revealing predicted synthesis rates that vary significantly among three different mechanisms modeled, including TJ (where the stalled and queuing ribosomes never terminate or dissociate from the mRNA; SAT (where only the stalled ribosome terminates and dissociates); and collision-stimulated abortive termination (CSAT, wherein termination of a stalled ribosome and dissociation occurs only on collision with the trailing elongating ribosome. The modeling predicts that increasing the initiation rate has no effect on YFP synthesis in the SAT model (where collisions by trailing ribosomes have no impact), but progressively reduces YFP synthesis in the TJ and CSAT models, where the effects of queuing ribosomes are paramount. The TJ and CSAT models also make predictions quite distinct from those of SAT when the number of stall sites increases, with SAT predicting much steeper and extensive declines, and the TJ and CSAT predictions differing in that additional stall sites have almost no impact for the TJ model. Increasing the distance between two stall sites is predicted to have no impact on YFP synthesis in the SAT model, to evoke a large step increase in synthesis in the CSAT models when the distance between stall sites is less than the ribosome footprint, and in the TJ model, YFP synthesis is predicted to exhibit a similar step increase followed by a gradual further increase that plateaus in the TJ model.

They went on to measure the effects on YFP reporter expression in response to altering the initiation rate with a single stall site, achieved by replacing the ATG start codon with near-cognates, or altering the Shine-Dalgarno sequence. They consistently observed the increased inhibitory effect with increasing initiation rate predicted by the TJ and CSAT models, and at odds with the SAT model, for various positions or sequences of a single stall site. Testing the effect of multiple stall sites in a variety of different configurations showed a better fit of the experimental data for the CSAT versus both the SAT and TJ models. The CSAT model also outperformed the other two models for the constructs where the distance between two stall sites was varied. They finish by presenting modeling results designed to show that the CSAT model out-performs the SAT model in explain in vivo observations of very low rates of premature termination from mRNAs lacking stall sites and the insensitivity of premature termination rates to overexpression of ribosome rescue factors; and the CSAT model outperforms the TJ model in explaining in vivo observations of very limited queuing of ribosomes upstream from stall sites. The authors conclude that collision-induced abortive termination is the most likely mechanism to account for the inhibitory effect of elongation stalling on translational output in *E. coli*.

General critique:

This study addresses an important problem, of how ribosome stalling during elongation reduces translation efficiency, and the approach taken to distinguish between the two leading models, of simple abortive termination (SAT) versus queuing of trailing ribosomes (TJ) back to the start codon, seems quite creative. Unfortunately, I can't judge the quality of the computational modeling. I also have doubts about whether the empirical data are good enough to rule out the two SAT and TJ models and argue strongly for the novel CSAT mechanism proposed here, as the discrimination between the 3 models shown in Figure 4 is generally more compelling than is the case for the related experiments shown in Figure 4—figure supplement 1, Figure 5—figure supplement 1 and Figure 6—figure supplement 1. In the end, I was persuaded that the results do significantly favor the CSAT model. However, I was unable to appreciate the significance or validity of the analyses presented in Figure 5, and feel that more effort is required to explain the logic of these modeling exercises and how they support the stated conclusions. In addition, more details are required concerning the experimental design and presentation of the data, as listed below.

Specific comments:

– Figure 1: in A, a better schematic and explanation of the differences between reporters 1-5 is required. In C, explain that 1 and 4 refer to the constructs as numbered in panel A.

– Figure 1: It would improve the experiment to detect the shorter stalled product expected for construct 5, to establish that the size of the detected product coincides with the position of the stall-inducing codon(s).

– Figure 2: use "little" k for rate constants; Figure 2: explain the arrows to the 3h species in the text or legend.

Figure 3 and associated text: More details are needed about exactly where the stall sites were inserted in panels A, B, C, which could also be depicted schematically.

– Figure 4 and Figure 4—figure supplement 1: explain the numbering system for the different codons examined, as their order in the sequence relative to the start codon.

Reviewer #2:

This report is written by Yitzhak Pilpel and Dvir Schirman

In this paper Ferrin and Subramaniam examine mechanisms of ribosome stalling on mRNA and their effects on expression level. They do so by introducing synthetic stall site into a fluorescent reporter and by examining the effect of number of stall sites, distance between adjacent stall sites, and initiation rate. They examine three different models that can explain the reduced expression level due to ribosome stalling, two previously proposed models, and a new model which suggests that collision of ribosomes at stalling sites causes abortive termination of the stalled ribosome.

Using simulation, the authors predict the behavior of protein synthesis rate for mRNA with stalling sites, under different regimes, and then they measure the synthesis rate of 100 different variants which are aimed to test the predictions of the simulations. By comparing the simulation predictions and the measured synthesis rate, they demonstrate that the collision associated model best explain the reduction in protein synthesis rate due to ribosome stalling.

This is a very interesting paper. The three models are clearly motivated and the combination of simulations with experiments on synthetic genes is certainly a sound one. The conclusions are of importance and of potential interest to the community, the experiments and simulations are done with rigor. In principle this paper could be suitable for *eLife*.

Yet, there are several points which we think that should be addressed before publication.

Major points:

1) The authors demonstrate the effect of ribosome stalling through a single stalling mechanism – "slow" codons of leucine under leucine starvation. It might be that the observations presented in this paper are through a different mechanism related specifically to leucine or to nucleotides in its codons, and not due to general ribosome stalling. We encourage the authors to test other means to stall ribosomes. One possibility can be starvation for a different amino acid. Perhaps easiest for the authors would be to examine their own synthetic genes from the PI's PNAS paper in which other yfp constructs for more amino acids were found to stall ribosomes upon starvation to additional amino acids.

Reviewer #3:

The manuscript is immediately interesting because it proposes a new and reasonable hypothesis for how ribosomal stalling might reduce protein expression - by abortive termination of a stalled ribosome caused by collision with an upstream elongating ribosome. The study argues for this novel model using a combination of experiments with synthetic YFP constructs in *E. coli*, under leucine starvation, combined with stochastic simulations of three alternative models.

Overall, whether or not the conclusions are entirely general or even correct, I believe the paper is important because it raises a novel and quite reasonable hypothesis. However, I have three major concerns about the paper in its current form, which can presumably be addressed in revision:

1) The study conclusions are limited to conditions of amino-acid starvation, but the authors discuss their results as if they hold generally;

2) The comparison between empirical data and three alternative models of ribosome stalling/abortion is done purely by visual inspection, without any statistical inference at all;

3) Some aspects of the model simulations contradict the experiments performed.

I will elaborate on these concerns below. But I want to re-iterate from the start that I remain quite enthusiastic about the work, for its conceptual novelty and admirable combination of experiment and theory.

Concern 1) The entire study (experiments and models) addresses the effects of ribosome stalling on decreased synthesis under a condition of severe amino-acid starvation. This is a perfectly reasonable biological question in its own right. But the authors must explain in their Abstract, Introduction, and Discussion that their results are limited to conditions of starvation, and they have no reason or evidence to favor one model over another for the effects of stalling on synthesis in healthy cells (indeed there may be almost no effect of a few stall sites on rates of synthesis in healthy *E. coli* cells). This needs to be clarified throughout the text, especially in the Abstract.

Concern 2) The authors consider three physical models: traffic jams from pause site all the way back to start codon (TJ), simple abortive termination (SAT), and collision-stimulated abortive termination (CSAT). Each of these models makes predictions according to stochastic simulations, with several fitted parameters; and the predictions are compared to empirical data in Figure 4. The authors conclude that CSAT matches the data best, but they do not seem to make any attempt at a statistical inference - that is to ask whether there is sufficient evidence to reject either of the classical models in favor of CSAT. Needless to say, such a statistical inference is de rigor in modern biological research, and the authors must actually perform statistics to draw their conclusions. It is not clear from visual inspection alone that CSAT does best in Figure 4, eg, the data are sometimes closer to SAT in 4C, and the data are sometimes closer to SAT in 4B as well. The point is that visual inspection is not a sound scientific basis for comparing prediction to data. Hence the entire field of statistics.

Fortunately, because they have a stochastic model, the authors could easily compute the likelihood of the empirical data under each model, and perform standard statistical inferences (eg likelihood ratio tests for nested models, or AIC for non-nested models). They should do so. Note that Figure 4 only shows the average synthesis rate under the models - averaged over a period of time equivalent to production of 10,000 control YFPs; the variance within this ensemble of simulations would allow the authors to compute the actual likelihood of the empirical data, as opposed to simply plotting average predictions over the experimental observations.

On a related note: the authors should state clearly how many parameters are fit, for each of the three models, for each of the experimental conditions? Are the same number of parameters fit for each model? This is a critical question that must be answered, as it pertains strongly to model comparison and inference.

Concern 3) The simulations are unrealistic in several respects, and not directly comparable to the experiments (to which they are compared in Figure 4). After reading the Methods in detail it appears that, for each model, simulations were run with a single control yfp mRNA as well as a single yfp sequence containing some combination of CTA (or other stalling) codons. This does not make much sense to me. Presumably, in the experiments, the ypf constructs are over-expressed in the *E. coli* cells, leading to depletion of available (free) ribosomes as a result of sequestration on the heterologous transcripts. And yet the simulation only considers two mRNA molecules in isolation, without any of the other native *E. coli* mRNAs? This is unrealistic, and it excludes the possibility that stalling may cause reduced protein expression due to a reduction of the pool of free ribosomes - a possibility that the model, as implemented, does not even consider. The authors should either simulate a realistic mRNA pool of native and heterologous transcripts, or explain this deficiency of the model openly - that they are not accounting for feedback between ribosomes stalled on the over-expressed construct and the pool of free ribosomes.

Moreover, it appear that each simulation considered the simultaneously translation of one wildtype yfp mRNA molecule along with a single modified yfp mRNA that contains some CTA codons. But the experiments surely express either the control construct OR the modified construct in any given cell. Why then do the simulations involve simultaneous translation of both types of mRNAs? (This may not be a problem because the authors are simulating an unrealistically small number of mRNAs of each species, reducing the possibility for feedback via the common pool of free ribosomes; but it is not a realistic setup compared to the experiments and it can easily be re-done properly). Do the authors see any feedback - ie, does the amount of time required to produce 10,000 control YFP proteins in the simulations depend on the identity of the other yfp transcript in those simulations? If not, then why simulate both construct simultaneously? And if so, then this underscores my previous point, that the authors should be simulating a realistic full complement of all mRNAs in the cell, including native ones and the heterologous ones.

[Editors' note: further revisions were requested prior to acceptance, as described below.]

Thank you for submitting your article "Kinetic modeling predicts a stimulatory role for ribosome collisions at elongation stall sites in bacteria" for consideration by *eLife*. Your article has been reviewed by one of the peer reviewers of the original manuscript, and the evaluation has been overseen by the Reviewing Editor, Alan Hinnebusch, and James Manley as the Senior Editor. The reviewers have opted to remain anonymous.

The reviewers found the revised version of your manuscript to be substantially improved and have accepted your rebuttal of their comments, with one exception. One of the reviewers remains unconvinced that the agreement between the data and the predictions of the competing models has been analyzed with the appropriate statistical methods, and has described what he/she thinks must be done to support the central claim of the paper in a rigorous, quantitative manner. We ask that you attempt to conduct the requested analysis and provide the results in revised version of the manuscript.

Reviewer #3:

I had three main concerns in my original review report:

1) Authors must state that conclusions are limited to stalling caused by aa starvation.

2) Authors must perform statistical comparison of three models (TJ, SAT, CSAT), not merely report mean square error.

3) Authors must perform simulations of the synthetic mRNAs in context of full complement of all other mRNAs (as opposed to in isolation).

The authors have addressed concern 1 through revised text. The authors do not perform the simulations suggested in concern 3, but argue that their transgenes are expressed at such low-levels that feedback on pool of available ribosomes is negligible. The authors offer additional experiments with integrated transgenes to support this. This is satisfying response. (Although it would still be better to perform the full simulations.)

With regard to point #2 (statistical comparison of models), the authors have not done much of anything in the way of new analysis, it seems. They do not provide any p-value to compare one model to another. They suggest that RMS is equivalent to reporting the Akaike Information Criterion. Even though all models being considered have the same number of parameters, they have not actually computed the likelihood under each model. The equivalence of AIC and RMS only occurs if the error distribution is the same under all models (eg all normal errors), but in fact the error distribution is surely not normal in these simulations. If the authors wish to use AIC for model comparison then they must actually compute the likelihood of the ML parameters under each model - which involves running an ensemble of simulations at each parameter to set to determine the likelihood of the data under that set of parameters. As it stands, the central claim comparing the TJ SAT and CSAT models rests on an entirely qualitative, non-statistical comparison.

Overall, the study makes a strong contribution. But I hesitate to offer acceptance until the data in support of the central claim have been analyzed statistically.

[Editors' note: further revisions were requested prior to acceptance, as described below.]

Thank you for resubmitting your work entitled "Kinetic modeling predicts a stimulatory role for ribosome collisions at elongation stall sites in bacteria" for further consideration at *eLife*. Your revised article has been favorably evaluated by James Manley as Senior editor, Alan Hinnebusch as Reviewing editor and one reviewer.

The manuscript has been improved, and both reviewer #3 and the reviewing editor are willing to recommend acceptance, but as you'll see in the review below, reviewer #3 still feels that there is a standard and superior way to provide statistical support in favor of one model versus the other, which you have not pursued. We wanted to give you one more chance to implement his/her suggestions before moving forward to publication, with the idea that you might be able to achieve a manuscript that would meet the most rigorous statistical standards. If you disagree with this reviewer's suggested approach, please provide us with a brief explanation for rejecting it and justifying the approach you have taken instead.

Reviewer #3:

My concern was that the authors did not provide statistical support in favor of one model versus another, given the data. There is a standard way to provide a statistical comparison of stochastic models - which I mentioned in my review, and which the authors have not addressed. What they should do is compute the most-likely parameters under each model, compute the actual likelihood of each model under the ML parameters, and then compare model fit using AIC. They have done nothing of the sort - indeed, they still fit the parameters by least squares (as opposed to maximum likelihood), and they never report the likelihood of the data under the inferred parameters, for any model.

In response to my critique they undertook extensive ensembles simulations of their models (with fixed parameters inferred by least squares) and they now compare the error distributions of the various models using AIC. Although this procedure seems correct, prima facie, it actually only makes sense if the parameters had been fit by maximum likelihood to begin with.

Overall, I think the paper should be published. The authors have made some attempt to provide statistical model comparison. Whatever deficiencies of this analysis remain, as long as the authors make their data available, anyone can analyze it and draw their own conclusions.

---

## [Author Response]

*Essential revisions:*

*It is necessary to respond to requests for revisions of text and figures to make the paper easier to read and understand.*

We have responded to these requests as explained below the specific points raised by the reviewers.

To address the key comment of Rev. #2, it is necessary to repeat a subset of the experiments with a second set of reporters containing a different codon besides leucine to rule out the possibility of effects specific to leucine starvation. One possibility would be to repeat only the analysis in Figure 4 for a different amino acid.

We have now performed additional experiments during serine starvation for measuring the effect of initiation rate (Figure 4—figure supplement 1) and the number of stall-inducing codons (Figure 5—figure supplement 1) on YFP synthesis rate.

In Figure 4—figure supplement 1, the stall-inducing TCG serine codon has a lower effect on YFP synthesis rate as the initiation rate of the *yfp* mRNA is decreased. This trend is similar to our observation during leucine starvation in Figure 4. This trend is predicted by both the CSAT and TJ models, but not the SAT model.

In Figure 5—figure supplement 1, we compared the effect of one vs. two TCG serine codons in *yfp* mRNA. The TJ model systematically overestimated YFP synthesis rate for 7 out of 9 double TCG *yfp* variants. Conversely, the SAT model underestimated the YFP synthesis rate for all 9 double TCG variants. The CSAT model did not show any such systematic bias and resulted in the smallest error between its predicted and measured YFP synthesis rates.

*To address the key comment of Rev. #3, it is important to acknowledge that the CSAT mechanism might not apply to ribosome stalling imposed by different mechanisms operating in non-amino acid starved cells.*

We have addressed this comment as follows:

1) The Abstract includes the sentence “To decipher ribosome kinetics at stall sites, we induced ribosome stalling at specific codons by starving the bacterium *Escherichia coli* for the cognate amino acid.”

2) The Introduction includes the sentence “While these conclusions are limited to the specific context of amino acid starvation in *E. coli*, the integrated approach developed in this work should be generally applicable to investigate other ribosome stalls in both bacteria and eukaryotes.”

3) The Discussion includes the sentence “Finally, we studied the CSAT model solely in the context of ribosome stalls caused by amino acid starvation in *E. coli*. Hence the validity of this model at ribosome stalls in exponentially growing bacterial cells remains to be tested.”

4) The first sentence in all the five result sections includes a reference to “amino acid starvation in *E. coli*”.

*Second, it is necessary to perform statistical analyses to determine whether the agreement of the experimental data with the CSAT model predictions is significantly better than that given by the SAT or TJ models.*

We have provided a root-mean square error estimate for each of our experiment-model comparisons (RMS error% in Figure 4, Figure 5, Figure 6). RMS error% does not consider the systematic bias in the predictions of the TJ and SAT models, and hence it is a conservative estimate of each model’s performance.

In each model, a single parameter — the unknown elongation rate at the stall site — is fit so that the model reproduces the measured YFP synthesis rate of reporters with single stalls and with ‘wild-type’ RBS. The RMS error% is calculated for the tested reporters, either with altered initiation rate or with multiple stalls. Since the same number of parameters are fit in each model, statistical inference measures such as Akaike Information Criterion (AIC) or Bayesian Information Criterion (BIC) will vary in proportion to the RMS error% (the contribution coming from number of fitted parameters is identical in all three models).

We have stated our fitting procedure in our main text for each experiment-model comparison (subsection “Experimental variables for distinguishing kinetic models of ribosome stalling”), in our Materials and methods (subsection “Simulation”), and provided our implementation code. All parameters used in our models are provided in Tables A–F.

*Third, you are asked to repeat the simulations to consider the more realistic situation where the reporter mRNAs with inserted Leu codons are being translated in the presence of a full complement of mRNAs and not merely a single control reporter mRNA lacking the Leu codon.*

We have addressed this concern through experiments and re-writing of our methods. We have included the reviewer’s full comment below to provide context for our subsequent explanation.

*Reviewer #1:*

[…]

*General critique:*

*This study addresses an important problem, of how ribosome stalling during elongation reduces translation efficiency, and the approach taken to distinguish between the two leading models, of simple abortive termination (SAT) versus queuing of trailing ribosomes (TJ) back to the start codon, seems quite creative. Unfortunately, I can't judge the quality of the computational modeling. I also have doubts about whether the empirical data are good enough to rule out the two SAT and TJ models and argue strongly for the novel CSAT mechanism proposed here, as the discrimination between the 3 models shown in Figure 4 is generally more compelling than is the case for the related experiments shown in Figure 4—figure supplement 1, Figure 5—figure supplement 1 and Figure 6—figure supplement 1. In the end, I was persuaded that the results do significantly favor the CSAT model. However, I was unable to appreciate the significance or validity of the analyses presented in Figure 5, and feel that more effort is required to explain the logic of these modeling exercises and how they support the stated conclusions. In addition, more details are required concerning the experimental design and presentation of the data, as listed below.*

The modeling exercises in Figure 7 (Figure 5 in original version) were performed to examine aspects of ribosome stalling that we did not investigate experimentally. Specifically, we examined two variables – the abortive termination rate and the ribosome density near stall sites – neither of which was directly measured in our work. First, the exact mechanism by which abortive termination occurs selectively at stalled ribosomes has remained unclear [Brandman and Hegde, 2016, Shao and Hegde, 2016]. Our modeling exercise in Figure 7 top panel shows that collision-stimulated abortive termination is naturally selective for stalled ribosomes. Second, our modeling exercise in Figure 7 bottom panel shows that the exact value of the abortive termination rate is not critical for robust abortive termination in the CSAT model. Finally, our Figure 7 shows that the predicted ribosome density near stall sites is in agreement with ribosome densities that have been measured previously using ribosome profiling. We have included this explanation in the main text accompanying Figure 7. We have also included a more speculative discussion of the consequences of our observations in Figure 7 as part of our Discussion section.

*Specific comments:*

*– Figure 1: in A, a better schematic and explanation of the differences between reporters 1-5 is required. In C, explain that 1 and 4 refer to the constructs as numbered in panel A.*

We have provided a better schematic indicating the location of all Leu codons, the length of the YFP reporter, and re-named the reporters with the prefix *‘yfp’* for clarity.

The caption of Figure 1 now includes the sentences *“Schematic of ribosome stalling reporters used in B–E. Blue vertical lines show the location of CTA Leu codons that cause ribosome stalling during Leu starvation in E. coli. Locations of CTG Leu codons that do not induce ribosome stalling are shown in grey.”*.

Each panel of Figure 1 also includes the reporter names introduced in Figure 1.

*– Figure 1: It would improve the experiment to detect the shorter stalled product expected for construct 5, to establish that the size of the detected product coincides with the position of the stall-inducing codon(s).*

We were unable to detect the shorter stalled product from Construct 5 (*yfp4* in revised version) using Western blotting. We presume that this is because these products were short (<8kD) and also had a range of sizes due to the presence of 7 closely spaced stall sites. We had intended Construct 5 primarily as a negative control for fluorescence and full-length YFP protein. Nevertheless, to address the reviewer’s concern, we performed an experiment with *yfp* reporters that had stall sites at two different locations. In this experiment, we also used truncated *yfp* reporters as size markers that had stop codons at the location of the stall sites. As shown in Figure 8, truncated products occur at exactly the size corresponding to the two different stall sites in the *yfp* reporters (*yfp5* and *yfp6*). We did not include this figure in our manuscript in order to have a consistent set of reporters across the different panels of Figure 1.

Author response image 1.Western blot to demonstrate abortive termination at stall-inducing codons during leucine starvation**DOI:**
http://dx.doi.org/10.7554/eLife.23629.019

*– Figure 2: use "little" k for rate constants; Figure 2: explain the arrows to the 3h species in the text or legend.*

Suggested changes incorporated in Figure 2 and in the caption of Figure 2.

*Figure 3 and associated text: More details are needed about exactly where the stall sites were inserted in panels A, B, C, which could also be depicted schematically.*

We have depicted the suggested schematic in Figure 3 and described it further in the associated caption.

*– Figure 4 and Figure 4—figure supplement 1: explain the numbering system for the different codons examined, as their order in the sequence relative to the start codon.*

The captions of Figure 4, Figure 5, Figure 6 (revised version) now mention "The Leu position(s) are labeled by their order of occurrence along yfp relative to the start codon (22 Leu codons total)."

The captions of Figure 4, Figure 5, Figure 6 (revised version) now mention "The Leu codon positions indicated in the schematic correspond to the following codon positions along yfp (with start codon: 1, stop codon: 239): 2: 15, 6: 46[…]".

*Reviewer #2:*

[…]

*Major points:*

*1) The authors demonstrate the effect of ribosome stalling through a single stalling mechanism – "slow" codons of leucine under leucine starvation. It might be that the observations presented in this paper are through a different mechanism related specifically to leucine or to nucleotides in its codons, and not due to general ribosome stalling. We encourage the authors to test other means to stall ribosomes. One possibility can be starvation for a different amino acid. Perhaps easiest for the authors would be to examine their own synthetic genes from the PI's PNAS paper in which other yfp constructs for more amino acids were found to stall ribosomes upon starvation to additional amino acids.*

We have addressed this concern as part of Essential revisions above.

*Reviewer #3:*

[…]

*Concern 1) The entire study (experiments and models) addresses the effects of ribosome stalling on decreased synthesis under a condition of severe amino-acid starvation. This is a perfectly reasonable biological question in its own right. But the authors must explain in their Abstract, Introduction, and Discussion that their results are limited to conditions of starvation, and they have no reason or evidence to favor one model over another for the effects of stalling on synthesis in healthy cells (indeed there may be almost no effect of a few stall sites on rates of synthesis in healthy E. coli cells). This needs to be clarified throughout the text, especially in the Abstract.*

We have addressed this concern as part of Essential revisions above.

*Concern 2) The authors consider three physical models: traffic jams from pause site all the way back to start codon (TJ), simple abortive termination (SAT), and collision-stimulated abortive termination (CSAT). Each of these models makes predictions according to stochastic simulations, with several fitted parameters; and the predictions are compared to empirical data in Figure 4. The authors conclude that CSAT matches the data best, but they do not seem to make any attempt at a statistical inference that is to ask whether there is sufficient evidence to reject either of the classical models in favor of CSAT. Needless to say, such a statistical inference is de rigor in modern biological research, and the authors must actually perform statistics to draw their conclusions. It is not clear from visual inspection alone that CSAT does best in Figure 4, eg, the data are sometimes closer to SAT in 4C, and the data are sometimes closer to SAT in 4B as well. The point is that visual inspection is not a sound scientific basis for comparing prediction to data. Hence the entire field of statistics.*

We have addressed this concern as part of Essential revisions above.

*Fortunately, because they have a stochastic model, the authors could easily compute the likelihood of the empirical data under each model, and perform standard statistical inferences (eg likelihood ratio tests for nested models, or AIC for non-nested models). They should do so. Note that Figure 4 only shows the average synthesis rate under the models averaged over a period of time equivalent to production of 10,000 control YFPs; the variance within this ensemble of simulations would allow the authors to compute the actual likelihood of the empirical data, as opposed to simply plotting average predictions over the experimental observations.*

*On a related note: the authors should state clearly how many parameters are fit, for each of the three models, for each of the experimental conditions? Are the same number of parameters fit for each model? This is a critical question that must be answered, as it pertains strongly to model comparison and inference.*

*Concern 3) The simulations are unrealistic in several respects, and not directly comparable to the experiments (to which they are compared in Figure 4). After reading the Methods in detail it appears that, for each model, simulations were run with a single control yfp mRNA as well as a single yfp sequence containing some combination of CTA (or other stalling) codons. This does not make much sense to me. Presumably, in the experiments, the ypf constructs are over-expressed in the E. coli cells, leading to depletion of available (free) ribosomes as a result of sequestration on the heterologous transcripts. And yet the simulation only considers two mRNA molecules in isolation, without any of the other native E. coli mRNAs? This is unrealistic, and it excludes the possibility that stalling may cause reduced protein expression due to a reduction of the pool of free ribosomes -- a possibility that the model, as implemented, does not even consider. The authors should either simulate a realistic mRNA pool of native and heterologous transcripts, or explain this deficiency of the model openly -- that they are not accounting for feedback between ribosomes stalled on the over-expressed construct and the pool of free ribosomes.*

Moreover, it appear that each simulation considered the simultaneously translation of one wildtype yfp mRNA molecule along with a single modified yfp mRNA that contains some CTA codons. But the experiments surely express either the control construct OR the modified construct in any given cell. Why then do the simulations involve simultaneous translation of both types of mRNAs? (This may not be a problem because the authors are simulating an unrealistically small number of mRNAs of each species, reducing the possibility for feedback via the common pool of free ribosomes; but it is not a realistic setup compared to the experiments and it can easily be re-done properly). Do the authors see any feedback - ie, does the amount of time required to produce 10,000 control YFP proteins in the simulations depend on the identity of the other yfp transcript in those simulations? If not, then why simulate both construct simultaneously? And if so, then this underscores my previous point, that the authors should be simulating a realistic full complement of all mRNAs in the cell, including native ones and the heterologous ones.

We agree with the reviewer’s concern that reporter over-expression will complicate the interpretation of our experiments by depleting the pool of free ribosomes. To avoid this complication, we had performed all our experiments using a very low copy *E. coli* plasmid vector (*SC*101 ori*). To unambiguously demonstrate that our results are not affected by reporter over-expression, we have now chromosomally-integrated a subset of our wild-type and stall-containing *yfp* mRNAs used in Figure 5 of our manuscript. As shown in Author-response Figure 2, YFP genes in our plasmid vectors are expressed at ~4-fold the level of the same gene present on the chromosome. As seen in Author-response Figure 3, the reduction in YFP expression caused by ribosome stalling during leucine starvation is essentially identical between our plasmid-borne and chromosomally integrated reporters, confirming that the copy number of our reporters does not influence our measurements.

Author response image 2.Comparison of measured YFP levels when expressed from chromo- some and plasmid during Leu-rich growth**DOI:**
http://dx.doi.org/10.7554/eLife.23629.020

Author response image 3.Comparison of measured YFP levels when expressed from chromo- some and plasmid during Leu starvation**DOI:**
http://dx.doi.org/10.7554/eLife.23629.021

We wish to highlight that our work includes a whole cell simulation of all native mRNAs that we had used for inferring the steady-state fraction of charged tRNAs during leucine starvation. As the reviewer has rightly pointed out, in the absence of reporter over-expression (and consequent depletion of free ribosomes), the exact reporter mRNA copy-number or the simultaneous presence of a second reporter does not affect our prediction for the effect of stall sites on YFP synthesis rate (relative to a no-stall control). In this regime where there is no feedback between the translation of native and heterologous mRNAs, we used the whole cell simulation to compute the steady state concentration of charged tRNAs in the cell (Run 1 in github code). We then used this whole-cell simulation results to simulate the simultaneous translation of a single mRNA molecule of the test reporter and the control reporter (Runs 3–5 in github code). Simulating just the reporters enables faster simulation times (few min vs. several hours for the whole-cell simulation) without any loss of accuracy. More importantly, since we measured only the YFP synthesis rate in this work, we believe that our reporter-only simulation with the minimal number of unobserved parameters provides a more transparent representation of the underlying kinetic model than an equivalent whole-cell simulation with thousands of unobserved parameters. Finally, the simultaneous translation of the two reporters is simply to ensure that we used exactly the same set of parameters for our test and control reporters during simulations and subsequent analyses. We regret that we had implicitly assumed much of this discussion to be superfluous in our earlier draft, and we have now included a summary of this explanation.

[Editors' note: further revisions were requested prior to acceptance, as described below.]

*Reviewer #3:*

*I had three main concerns in my original review report:*

*1) Authors must state that conclusions are limited to stalling caused by aa starvation.*

*2) Authors must perform statistical comparison of three models (TJ, SAT, CSAT), not merely report mean square error.*

*3) Authors must perform simulations of the synthetic mRNAs in context of full complement of all other mRNAs (as opposed to in isolation).*

*The authors have addressed concern 1 through revised text. The authors do not perform the simulations suggested in concern 3, but argue that their transgenes are expressed at such low-levels that feedback on pool of available ribosomes is negligible. The authors offer additional experiments with integrated transgenes to support this. This is satisfying response. (Although it would still be better to perform the full simulations.)*

*With regard to point #2 (statistical comparison of models), the authors have not done much of anything in the way of new analysis, it seems. They do not provide any p-value to compare one model to another. They suggest that RMS is equivalent to reporting the Akaike Information Criterion. Even though all models being considered have the same number of parameters, they have not actually computed the likelihood under each model. The equivalence of AIC and RMS only occurs if the error distribution is the same under all models (eg all normal errors), but in fact the error distribution is surely not normal in these simulations. If the authors wish to use AIC for model comparison then they must actually compute the likelihood of the ML parameters under each model – which involves running an ensemble of simulations at each parameter to set to determine the likelihood of the data under that set of parameters. As it stands, the central claim comparing the TJ SAT and CSAT models rests on an entirely qualitative, non-statistical comparison.*

*Overall, the study makes a strong contribution. But I hesitate to offer acceptance until the data in support of the central claim have been analyzed statistically.*

We thank the reviewer for the comments regarding the distribution of errors. To clarify our statistical analysis, we note that there are three main sources of error/uncertainty in our work:

1) The first source is from imperfect simulation of the underlying kinetic model. We refer to this as simulation error below. We assume that this is the error the reviewer has in mind when referring to non-normal distribution of errors.

2) The second source is from imperfect measurement of YFP synthesis rate for any given yfp mutant. We refer to this as measurement error below. This error is represented by the error bars in Figure 4, Figure 5 and Figure 6.

3) The third source is from imperfect kinetic modeling, i.e. none of our models fully account for all the molecular processes that affect the measured YFP synthesis rate. We refer to this as modeling error below. This modeling error could either be relevant to our work, for eg. Inclusion / exclusion of abortive termination; or it might not be of interest for the question at hand, for eg. Local sequence context effects that are specific to individual yfp mutants.

In order to perform statistical inference, we consider each of these errors below:

1.1) Simulation Error

Before we consider the error in our simulation, we clarify the definition of YFP synthesis rate in our simulations. In each simulation run, YFP synthesis rate is defined as the number of YFP molecules produced from a yfp mutant during the time it takes to produce 10,000 YFP molecules from the no-stall control, normalized by 10000. We chose this definition of YFP synthesis rate in our simulations to match the number of YFP molecules that we expect to be produced in a cell during the course of our experimental measurement over 1-2 hours, based on a typical translation initiation rate of ~0.3 s^−1^ and a typical mRNA copy number of 10 molecules per cell (for a chromosomally-integrated gene expressed from a moderately strong promoter).

To address the reviewer’s suggestion to consider the error in a ensemble of identical simulations, we performed 5,000 independent and identically-parametrized simulations for each of the three kinetic models considered in our work. We show the distribution of YFP synthesis rate across this simulation ensemble for a double stall-site mutant in Figure 11 using parameters that were inferred from the corresponding single-mutant measurements. We note that the width of the distribution of simulation errors is approximately the standard error of our YFP synthesis rate estimate and is close to normal. Notably, these simulation errors are much smaller than the modeling errors discussed below (compare X-axis in Figure 11 and Figure 12).

1.2) Measurement Error

We performed all our measurements of YFP synthesis rate in triplicates from three clonal *E. coli* colonies. The error bars shown in Figure 4, Figure 5, and 6 reflect the standard error of measurement over these triplicates. We show the distribution of this measurement error in Figure 12. As seen in this figure, measurement errors are reasonably approximated by a normal distribution and are smaller than the modeling errors discussed below (compare X-axis in Figure 12 and Figure 13).

Author response image 4.Distribution of simulation error.Simulation error was calculated as the deviation of the YFP synthesis rate from the mean across an ensemble of 5000 identically simulations for each of the three kinetic models considered in the manuscript. The *yfp* mutant in the simulation had two CTA stall sites at Leu2 and Leu14 as shown in Figure 5. YFP synthesis rate was calculated relative to the no-stall control. Black line indicates a Gaussian fit.**DOI:**
http://dx.doi.org/10.7554/eLife.23629.022

Author response image 5.Distribution of measurement error.Measurement error was calculated as the standard error of the mean measured YFP synthesis rate for all the *yfp* mutants used in our manuscript (N=115). We note that even though standard error is strictly positive, we have duplicated and inverted the duplicated part in the definition of measurement error for ease of comparison with other errors. YFP synthesis rate was calculated relative to the no-stall control. Black line indicates a Gaussian fit.**DOI:**
http://dx.doi.org/10.7554/eLife.23629.023

1.3) Modeling Error

The residual difference between the experimental measurements of YFP synthesis rate and the corresponding prediction from each of the three kinetic models is shown in Figure 13. Notably, this error is much larger in magnitude than the simulation error in Figure 11.

From the distribution of modeling errors, we can make the following statistical inferences: Using the non-parametric one-sided Mann-Whitney test, we can conclude that the SAT model systematically underestimates the measured YFP synthesis rate (P=0.01, N=94) while the TJ model systematically overestimates the measured YFP synthesis rate (P=0.05, N=94). By contrast, the CSAT model’s prediction shows no such overestimation or underestimation bias (P > 0.45, N=94 for both cases). If we assume normal distribution of modeling errors (see discussion below) and use the one-sided Student’s t test, we can then conclude that the SAT model systematically underestimates the measured YFP synthesis rate (P < 10^−8^) while the TJ model systematically overestimates the YFP synthesis rate (P < 10^−15^). The CSAT model shows no such bias (P > 0.7).

Author response image 6.Distribution of modeling error.Modeling error was calculated as the residual difference between the mean prediction for YFP synthesis rate from each of the three kinetic models and the measured YFP synthesis rate for the corresponding *yfp* mutant (N=94). YFP synthesis rate for both the modeling prediction and the measured value was calculated relative to the no-stall control. Black line indicates a Gaussian fit.**DOI:**
http://dx.doi.org/10.7554/eLife.23629.024

The residual difference between our prediction and experiments (Figure 13) is much larger than the simulation error (Figure 11). This suggests that additional stochastic processes have acted on the measured YFP synthesis rate that have not been included in our modeling. Since our goal is only to distinguish between the three kinetic models considered in this work, we account for this unexplained variability outside of the stochastic simulation, following suggested practice (Hartig et al., 2011). Specifically, we assume that additional variability is independently and normal distributed. We believe that this assumption is the most parsimonious one based on our expectation that the additional variability arises from local sequence-context effects that are specific to each yfp mutant. Additive combination of several independent mutant-specific effects will result in a normal distribution of residuals. This assumption is consistent with our observation that the empirical residual distribution for the CSAT model (which has no prediction bias) is reasonably approximated by a normal distribution (black line in left panel in Figure 13). This practice of using the empirical residuals to test the assumption of normal error distribution has been used previously in diverse biological contexts where the underlying models do not capture the full variability present in the measured data (Martínez et al., 2011; Coulon et al., 2014; Wood et al., 2012

We then use the Akaike Information Criterion (AIC) for obtaining the statistical evidence favoring each of the three models considered in our work. AIC is an estimate of the expected, relative distance (Kullback-Leibler distance) between the fitted model and the unknown true mechanism that actually generated the observed data (Burnham and Anderson, 2013). The model with the lowest AIC is considered the best model since it will minimize the distance between the model prediction and observed data. Under the assumption of an external error model for the residuals with normal distribution and unknown constant variance, AIC can be computed from the empirical residuals as (Burnham and Anderson, 2013):AIC=nlogσ2+2K

where σ2 is the ML estimate of the residual variance:σ2=∑ϵ2in

and ϵi are the estimated residuals for the candidate model. *n* is the number of observations. *K* is the total number of fitted parameters including the unknown σ2. In our case, K=1 for all three models since the unknown stall strength has been fit using only the single mutant data, while the residuals are computed only for the initation rate mutants and multiple stall site mutants.

The Akaike weight in favor of model i can then be calculated as:wi=exp(−Δi/2)∑jexp(−Δj/2)

where the sum over j runs over the three models TJ, SAT, and CSAT considered in this work. Δi=AICi−AICmin where AICmin corresponds to the model with the lowest AIC value.

In our case, the measurement error is the same for all three models and the simulation error is negligible compared to the modeling error. Hence we can apply the Akaike weight formula (3) to our modeling errors and arrive at the following weights in favor of each model:

Model∆_i_Akaike weightTJ137<0.001SAT130< 0.001CSAT0> 0.999

We have summarized the statistical inference results for the Student’s t -test and Akaike weights in our Results section.

[Editors' note: further revisions were requested prior to acceptance, as described below.]

*Reviewer #3:*

*My concern was that the authors did not provide statistical support in favor of one model versus another, given the data. […] Whatever deficiencies of this analysis remain, as long as the authors make their data available, anyone can analyze it and draw their own conclusions.*

We understand Reviewer #3’s suggested approach for statistical comparison of stochastic models. However, we believe that it is inapplicable to our work because the stochasticity in our simulations reflects only the error in our simulations, and not the error in our modeling. As we highlighted in our previous round of response, the simulation error is negligible in comparison to our modeling error. In other words, our simulations are essentially perfect and deterministic representations of our kinetic models. If we follow the procedure suggested by Reviewer #3, the likelihood of each model given the data will be zero since none of our simulation runs will reproduce the measurements for most of the YFP mutants.

Since the processes that cause the modeling error are not of interest to us (for eg. local sequence context effects), we have followed a standard approach [Hartig et al., Ecology Letters, (2011) 14:816–827 Statistical inference for stochastic simulation models – theory and application] of accounting for the modeling error outside of the stochastic simulation. Our procedure follows the prescription that has been suggested by Hartig et al.: “If it is difficult to specify an explicit statistical error model from the data, informal likelihoods offer an alternative. By informal likelihoods, we understand any metric that quantifies the distance between the predictions of a stochastic simulation model and the observed data, but is not immediately interpretable as originating from an underlying stochastic process (see Beven 2006; Smith et al., 2008). A common example is the sum of the squared distances between Sobs and the mean of Ssim (Refsgaard et al., 2007; Winkler & Heinken 2007)”.

Finally, we note that the Akaike weights for each model that we have reported are just normalized likelihood values (Burnham and Anderson, Springer (2002); Model Selection and Multimodel Inference).